SOFTWARE

# PyFibers: An open-source NEURON-Python package to simulate responses of model nerve fibers to electrical stimulation

Daniel P. Marshall[1], Elie S. Farah[1,2], Eric D. Musselman[1], Nicole A. Pelot[1]*, Warren M. Grill[1,3,4,5]*

1 Department of Biomedical Engineering, Duke University, Durham, North Carolina, United States of America, 2 Department of Computer Science, Duke University, Durham, North Carolina, United States of America, 3 Department of Electrical and Computer Engineering, Duke University, Durham, North Carolina, United States of America, 4 Department of Neurobiology, Duke University, Durham, North Carolina, United States of America, 5 Department of Neurosurgery, Duke University, Durham, North Carolina, United States of America

* warren.grill@duke.edu (WMG); nikki.pelot@duke.edu (NAP)

## Abstract

Computational modeling of peripheral nerve fibers is a key tool for designing improved neuromodulation therapies. The NEURON software is commonly used to create biophysical simulations of nerve fibers, often in the outdated HOC language. Whether written in HOC or Python, implementing fiber simulations involves a steep learning curve and requires a large amount of standard, boilerplate code that is typically written anew for each project. There is a need for a code package that standardizes and simplifies the creation of model fibers, the execution of simulations of electrical stimulation, and the analysis of the resulting data. We created PyFibers, a NEURON-Python package that provides tools for accomplishing all these tasks and supports the development of new fiber models and stimulation protocols. PyFibers includes 11 fiber models from prior publications under a shared framework, and we validated each model's implementation in PyFibers against the original results. Our open-source tool simplifies and standardizes computational modeling of peripheral nerve fiber responses to electrical stimulation, providing a platform for the development of novel therapies using electrical stimulation, block, and recording.

## Author summary

Electrical stimulation of peripheral nerves can treat conditions such as epilepsy, paralysis, bladder dysfunction, and sleep apnea. Improving the design of these nerve stimulation therapies can increase efficacy and reduce side effects. Computational models of nerve fibers (axons) provide a rapid approach to study many different device designs and parameters. However, present nerve

**Data availability statement:** PyFibers is open-source and publicly available from PyPI https://pypi.org/project/pyfibers/ and GitHub https://github.com/wmglab-duke/pyfibers. The documentation is hosted on GitHub at https://wmglab-duke.github.io/pyfibers/. Specific versions of the PyFibers code can be cited using Zenodo DOIs from https://zenodo.org/records/17178184. The data presented in this paper along with all code to recreate the figures are available from sparc.science at https://doi.org/10.26275/8ssx-gcil.

**Funding:** This work was supported by the National Institutes of Health (https://www.nih.gov/) (75N98022C00018 to NAP and WMG; OT2 OD025340 to WMG and NAP; R01 NS126376 to WMG). The funders had no role in study design, data collection and analysis, decision to publish, or preparation of the manuscript.

**Competing interests:** The authors have declared that no competing interests exist.

fiber modeling tools focus primarily on modeling of entire nerves, and researchers often need to create and run nerve fiber models on their own. We created PyFibers, an open-source Python package that makes it easy to implement and simulate computational models of stimulation, block, and recording of single nerve fibers. With PyFibers, users can quickly create nerve fiber models, apply electrical stimulation, and examine the responses in detail. PyFibers runs on top of a widely used neuron modeling program (NEURON) and can plug into larger simulation pipelines, enabling researchers to test new electrode designs and stimulation strategies to complement preclinical or clinical studies.

## Introduction

Electrical stimulation of the peripheral nervous system is used to treat a broad range of diseases and disorders. For example, the FDA has approved vagus nerve stimulation to treat epilepsy [1], to treat depression [2], and as an adjunct to stroke rehabilitation [3]; sacral nerve stimulation to treat urgency frequency syndrome, urinary retention, and urge incontinence [4,5]; and hypoglossal nerve stimulation to treat obstructive sleep apnea [6]. While such therapies are clinically effective, the mechanisms of action for many applications of peripheral nerve stimulation remain unclear, and optimal stimulation parameters and electrode designs are not established. Computational models enable rapid iteration of device designs and stimulation parameters; further, anatomically and biophysically realistic models can provide mechanistic insights into the responses to stimulation.

The NEURON simulation environment [7] is a robust and widely used platform for modeling realistic neurons, including their responses to electrical stimulation. Neuronal responses to stimulation can be solved numerically using the cable equation and differential equations that describe the dynamics of voltage-gated ion channels. NEURON models are typically developed in the HOC programming language; HOC—sharing much of its syntax with C—has few users outside of the NEURON community and presents a steep learning curve for new users. NEURON introduced support for scripting using Python in 2009 [8]. In contrast to HOC, Python is much more accessible and widely adopted in academia and industry [9]. Even when NEURON simulations are implemented in Python, substantial barriers to widespread use remain. Simulations require a great deal of boilerplate code, and models are often difficult to reproduce due to incorrect or incomplete descriptions provided in publications [10].

As peripheral nerve stimulation becomes more prevalent in clinical settings and research applications [11], the importance of robust, reproducible, and accessible modeling tools is increasing [12]. No existing open-source solutions are dedicated to computational modeling of electrical stimulation of peripheral nerve fibers; rather, previous publications focused on multi-scale modeling of peripheral nerves [13–16]. Insufficient focus on nerve fiber modeling has left present solutions lacking in several areas (Table 1). An open-source, standardized, user-friendly package for modeling

**Table 1. Comparison between PyFibers and other published open-source tools that include computational modeling of nerve fiber responses to electrical stimulation.**

| Features | PyFibers | NRV [13] | VINERS [14] | ASCENT [16] | PyPNS [15] |
|---|---|---|---|---|---|
| **Model nerve fiber API** | Yes | Yes | No | No | No |
| **Dependencies** | Python | Python Gmsh FEniCS | MATLAB Gmsh EIDORS | Python Java COMSOL | Python |
| **Commercial software required** | No | No | Yes | Yes | No |
| **Field modeling** | Internal/External | Internal/External | Internal | Internal | Internal/External |
| **Nerve geometry modeling** | External | Internal/External | Internal | Internal | Internal/External |
| **Included fiber models** | dMRG | | dMRG | dMRG | dMRG |
| | iMRG | iMRG | | iMRG | |
| | Peña | | | Peña | |
| | Sweeney | | | | |
| | | Gaines | Gaines | | |
| | Rattay | Rattay | | Rattay | |
| | Sundt | Sundt | Sundt | Sundt | |
| | Tigerholm | Tigerholm | | Tigerholm | |
| | Schild97 | Schild97 | | Schild97 | |
| | Schild94 | Schild94 | | Schild94 | |
| | Thio Cutaneous | | | | |
| | Thio Autonomic | | | | |
| | | HH | | | HH |
| **Interface for user-developed fiber models** | Yes | Requires Implementation | No | No | No |
| **Interface for user-developed simulation code** | Yes | No | No | No | No |
| **Support for 3D fibers** | Yes | No | No | No | No |
| **Support for plugins** | Yes | No | No | No | No |

All solutions listed depend on NEURON for neuronal simulation. "Internal" and "External" refer to whether the feature is built into the software or relies upon inputs from a separate software package. HH = Hodgkin Huxley, MRG = McIntyre-Richardson-Grill, dMRG = MRG-discrete which allows only specific fiber diameters, iMRG = MRG-interpolation which is an interpolation of the discrete formulation. Note that some platforms have different implementations of these concepts. For example, NRV and ASCENT use two different interpolations of the original MRG model. A detailed treatment of the fiber models available in PyFibers is provided in Table 2.

stimulation of nerve fibers would lower barriers to entry and reduce duplication of efforts, thereby easing implementation and reuse of fiber models, reducing errors, and improving rigor and replicability.

To address these needs, we created PyFibers, an open-source Python package for defining model nerve fibers and simulating their responses to electrical stimulation in NEURON. PyFibers includes 11 previously published fiber models, provides extensive control over model parameterization, and enables thorough access to simulation data. PyFibers' object-oriented design promotes modularity and extensibility: users can readily develop new fiber models and simulation protocols, as well as integrate PyFibers into larger nerve modeling workflows; for example, we replaced the HOC code that previously comprised the fiber simulation backend of ASCENT [16] with PyFibers, enabling a substantial reduction in code complexity. Model nerve fibers in PyFibers can be stimulated using electrical potentials generated by existing peripheral nerve modeling software [13–16], custom field models (e.g., finite element models in COMSOL), analytical calculation with built-in functions for extracellular point sources, or intracellular current sources. Therefore, PyFibers is a simulation infrastructure that streamlines researcher development and use of fiber modeling.

In this publication, we detail the design and implementation of PyFibers, we provide example use cases, and we provide guidelines to help ensure numerical accuracy and correct model parameterization. We highlight how PyFibers streamlines the process of implementing model fibers and stimulation paradigms, reducing coding burden and complexity. The PyFibers codebase is complemented by thorough documentation, tutorials, and unit testing to ensure rigor. After creating PyFibers, we iterated with alpha testers from our research group and beta testers from other research groups, and we modified the documentation and interface to improve the user experience. Overall, PyFibers represents a significant advance in modeling electrical stimulation of peripheral nerve fibers, an increasingly important component in the design and optimization of neuromodulation therapies.

## Design and implementation

PyFibers uses the Python implementation of NEURON [7,8]. Simple installation via PyPI, along with extensive documentation and tutorials, facilitate user adoption. We tested PyFibers on Windows, Linux, and macOS. At release, PyFibers supports Python 3.10–3.13 and NEURON 8 and 9. For the data generated herein, we used Python 3.11 with NEURON 8.2.6. Our unit tests are implemented with tox (https://tox.wiki/en), which makes testing compatibility with new NEURON and Python versions straightforward. In this section, we describe basic use and implementation of PyFibers.

### Operation

The use of PyFibers is detailed in our online documentation (https://wmglab-duke.github.io/pyfibers) and briefly described herein. PyFibers can be installed from the Python Package Index (PyPI):

```
pip install pyfibers
```

After installation, users must compile the NEURON NMODL mechanisms:

```
pyfibers_compile
```

An example workflow to simulate extracellular stimulation is outlined in Fig 1 and detailed in Box 1. This example is representative rather than an exhaustive demonstration of PyFibers' features. Further demonstrations of simulations leveraging PyFibers are provided in the "Use cases" section.

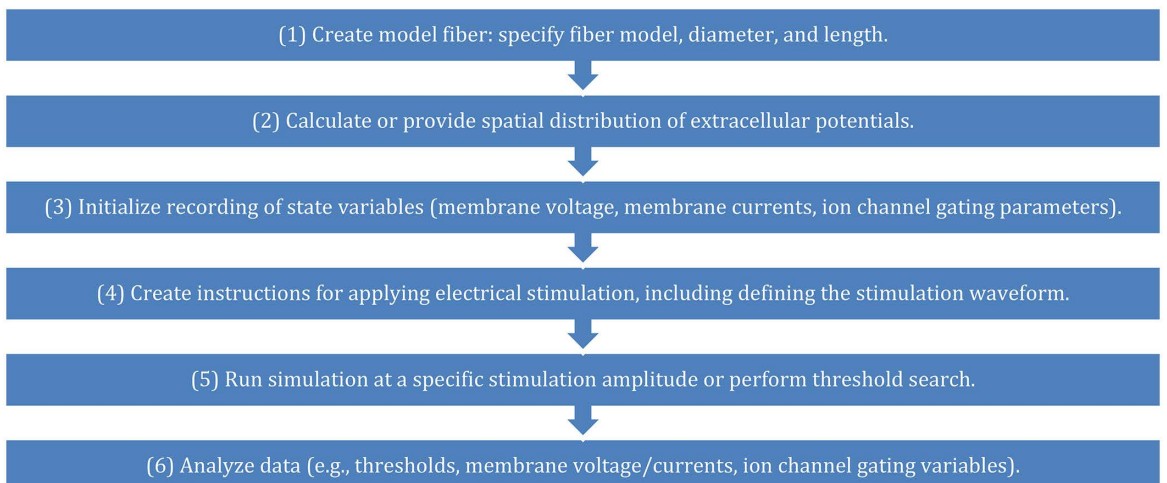

**Fig 1. Example sequence of operations in PyFibers to create a model fiber and simulate its response to extracellular stimulation.**

## Box 1. Example PyFibers simulation using the steps defined in Fig 1.

(1) **Fiber creation:** The user creates a non-branching model fiber by specifying—at minimum—the fiber model (ultrastructure and channels), diameter (µm), and length (number of nodes, number of sections, or microns).

```
from pyfibers import build_fiber # Enables creation of a model fiber
from pyfibers import FiberModel # Defines the available fiber models
fiber = build_fiber(fiber_model=FiberModel.MODEL, diameter=float, length=float)
```

(2) **Extracellular potentials**: If simulating extracellular stimulation, electrical potentials at the center of each section of the model fiber are required. Typically, a current of i0=1 mA is used to represent a unit stimulus, which can later be scaled to any arbitrary stimulation amplitude. Extracellular potentials can be calculated analytically from one or more point current sources (located at x, y, z [µm], noting that fibers are created at x=0, y=0, and extend in the positive z direction by default) in an infinite, homogeneous medium with isotropic (float) or anisotropic (tuple of ($\sigma_x$, $\sigma_y$, $\sigma_z$)) conductivity [S/m]. Alternatively, users can use externally calculated potentials (e.g., from a finite element model, described later).

```
# Calculate spatial distribution of extracellular potentials from a point current source at
each fiber section and attach to fiber
fiber.potentials = fiber.point_source_potentials(x=float, y=float, z=float, i0=float,
sigma=float | tuple(float))
```

(3) **State variable recording**: The user can enable saving of state variables of the model fiber, including the transmembrane potential, the gating parameters, and/or the transmembrane currents. By default, these functions save state variables for all nodes (i.e., excitable sections of the fiber, see "Implementation: fiber.py: Implementing and controlling model fibers" for details) at every time step during a simulation (step #5).

```
fiber.record_vm()        # Save transmembrane voltage for subsequent runs
fiber.record_gating()    # Save gating parameters for subsequent runs
fiber.record_im()        # Save transmembrane current for subsequent runs
```

(4) **ScaledStim class and stimulation waveform**: The user defines a unit stimulation waveform that scales the extracellular potentials from step #2 for each time step. The waveform is defined as a `Callable` (e.g., a function) that takes a single time value as a float input (in milliseconds) and returns a single waveform value as a float. In this example, the waveform is a 0.15 ms square pulse from t = 0.1 to t = 0.25 ms. The user can write a custom function to define a waveform or use interpolators (e.g., from numpy or scipy).

```
start, on, off, stop = 0, 0.1, 0.25, 50 # in ms

# Create function which takes in a time and returns a waveform value
def my_square_pulse(t: float) -> float:
    if t>=on and t<off:     # if time is during the pulse, in ms
        return 1 # on
    else:                   # if time is outside the pulse, in ms
        return 0 # off

# Same waveform as above, but using scipy.interpolate.interp1d
my_square_pulse = interp1d(
    [start, on, off, stop], # times in ms
    [0, 1, 0, 0],           # waveform values at those times
    kind="previous"
)
```

The user then initializes a `ScaledStim` class instance, which defines a set of instructions for executing a simulation of extracellular stimulation. This requires providing a waveform, e.g., one of the examples of `my_square_pulse()` defined above:

```
from pyfibers import ScaledStim
stimulation = ScaledStim(waveform=Callable, dt=float, tstop=float)
```

(5) **Simulation**: The user simulates the fiber response to stimulation with either a specific current amplitude (amp):

```
# Run fiber simulation with ScaledStim
ap, time = stimulation.run_sim(amp=float, fiber=fiber)
```

or a bisection search to identify activation (or block) threshold:

```
# Run a search for activation threshold
amp, ap = stimulation.find_threshold(fiber, condition="activation")
```

(6) **Data analysis**: After the simulation is complete, the user can access and analyze the saved data (if the user performed step 3 above) from the most recent simulation:

```
time                  = stimulation.time  # Fig 2A-2D
vm_over_time          = fiber.vm          # Fig 2A, 2B, 2D
gating_vars           = fiber.gating      # Fig 2C
i_membrane_over_time  = fiber.im          # Not demonstrated in Fig 2
```

The user can also perform other analyses, such as creating a recording of a single fiber action potential (SFAP). This requires electrical potentials from a recording electrode, representing the sensitivity of the recording electrode to extracellular currents along the fiber (Peña et al., 2024).

```
# Calculate electrical potentials from a point source electrode
recording_sensitivity = fiber.point_source_potentials(x=float, y=float, z=float, i0=float,
sigma=float | tuple(float))

# Record the single fiber action potential
sfap = fiber.record_sfap(recording_sensitivity) # Fig 2D
```

Thus, in the simplest use case, PyFibers reduces the implementation and stimulation of a model fiber with extracellular stimulation to five lines of code (Box 2 and Fig 2). For comparison, an equivalent simulation requires >600 lines of HOC or Python code without PyFibers (From ASCENT v1.0.0 [26]: CreateAxon_Myel.hoc, GeometryBuilder.hoc, RunSim.hoc, and FindThresh.hoc together constitute 668 lines of code). This complexity poses a barrier to entry by increasing the initial effort required to learn and script simulations, and it is detrimental to rigor by creating more opportunities for errors in the implementation. In addition, such manually scripted approaches lack the wealth of simulation options, built-in fiber models, and analysis tools provided by PyFibers.

---

**Box 2. Introductory example of PyFibers code to implement a 10 μm diameter myelinated fiber (MRG-interpolation, 25 nodes = 27 mm in length) and simulate its response to extracellular stimulation. The extracellular potentials are from a point source of current located halfway along the fiber at an electrode-fiber distance of 250 μm in an isotropic, homogeneous medium with a conductivity of 10 S/m. The stimulation waveform is a monophasic cathodic rectangular pulse with pulse duration of 0.15 ms (from 0.1 to 0.25 ms). A ScaledStim instance is created and provided with a waveform, and the current amplitude corresponding to the activation threshold is then determined. Results are shown in Fig 2.**

```
fiber            = build_fiber(FiberModel.MRG_INTERPOLATION, diameter=10, n_nodes=25)
fiber.potentials = fiber.point_source_potentials(x=0, y=250, z=fiber.length / 2, i0=1,
sigma=10)
waveform         = interp1d([0,0.1,0.25,50], [0,1,0,0], kind="previous")
stimulation      = ScaledStim(waveform=waveform, dt=0.001, tstop=50)
amp, ap          = stimulation.find_threshold(fiber, condition="activation")
```

---

Additional examples of workflows leveraging PyFibers are demonstrated in the "Use cases" section, including modeling kilohertz frequency block, modifying fiber geometry, using PyFibers within a nerve modeling pipeline to simulate activation and recording, and implementing a new fiber model. Further information on the operation of PyFibers is given in the API documentation (https://wmglab-duke.github.io/pyfibers/autodoc/index.html). The PyFibers documentation also includes detailed tutorials (https://wmglab-duke.github.io/pyfibers/tutorials/index.html) for the following:

• Basic tasks

  • Running a simulation for a single stimulation amplitude

  • Determining activation thresholds

  • Analyzing simulation results

  • Generating recorded signals (single fiber action potentials)

• Complex tasks

  • Resampling high-resolution potentials from an external potential source

  • Determining block thresholds

  • Running multiple simulations in parallel

  • Creating fibers using 3D paths

  • Simulating stimulation from multiple current sources

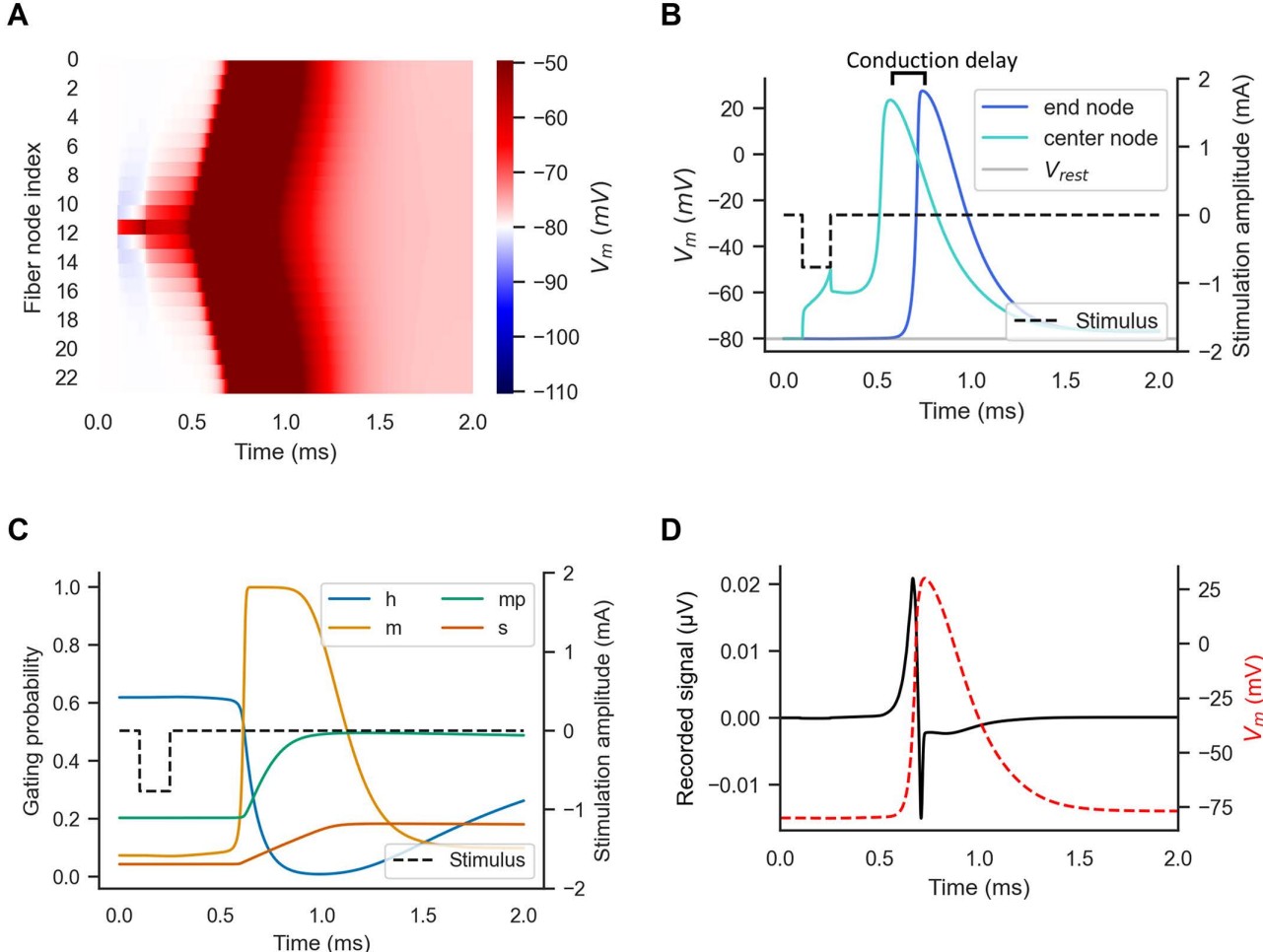

**Fig 2. Simulation results generated by the code in Box 2; code for recording state variables, calculating the single fiber action potential, and generating plots is not included in Box 2, but can be found in the dataset associated with this manuscript.** Stimulation of a 10 µm diameter MRG fiber with a length of 25 nodes (27 mm). The extracellular potentials were calculated using a point source of current located halfway along the fiber at an electrode-fiber distance of 250 µm in an isotropic, homogeneous medium with a conductivity of 10 S/m. The stimulation was a 0.15 ms, monophasic rectangular pulse (from t = 0.1 to 0.25 ms) delivered at threshold amplitude (−0.766 mA). A) Transmembrane potential plotted for all fiber nodes. $V_m = -80$ mV (white) is the resting transmembrane potential. B) Transmembrane potential recorded at node 11 ("center") and node 23 ("end", i.e., second-to-last since the end nodes are passive). C) Gating variables for the center node. D) Single fiber action potential recorded from a point electrode at 90% of the fiber length with a 250 µm electrode-fiber distance in an isotropic medium with conductivity of 10 S/m. $V_m$ = transmembrane voltage (mV).

## Implementation

PyFibers is structured around two primary modules. The `fiber.py` module enables the creation and manipulation of model nerve fibers and provides tools for users to develop custom fiber models. The `stimulation.py` module calculates fiber responses to electrical stimulation and provides tools for developing custom simulation protocols. Additionally, the `models/` directory houses a Python module for each of the 11 built-in fiber models. This section describes the purpose, design, and functionality of each module, each of which is detailed in our API documentation (https://wmglab-duke.github.io/pyfibers/autodoc/index.html). The discussion of edge cases and error handling is non-exhaustive; users should be mindful of warnings and errors thrown by PyFibers and are ultimately responsible for appropriate model design and usage.

**fiber.py: Implementing and controlling model fibers.** The `fiber.py` module defines a generalized `Fiber` class to create and modify model fibers. The module's `build_fiber()` function returns a Fiber class instance (i.e., model fiber) consisting of a series of connected NEURON sections with the ultrastructure, electrical properties, and ion channel mechanisms of the user-selected fiber model and user-defined fiber diameter and length. Full descriptions of parameters for creating model fibers are found in our documentation of the fiber module (https://wmglab-duke.github.io/pyfibers/autodoc/fiber.html).

**Fiber models:** Each module (i.e., `.py` file) in the `models/` directory contains a single class—built as a subclass of the `Fiber` class—that describes the mechanisms and instructions for a specific fiber model. For details, see the documentation on available fiber models (https://wmglab-duke.github.io/pyfibers/fiber_models.html). PyFibers includes implementations of 4 myelinated and 7 unmyelinated fiber models (Table 2). In a later section ("Validation against previously published nerve fiber model implementations"), we simulate responses for each fiber model to ensure that the PyFibers implementations replicate the published results.

Appropriate choice of fiber model (i.e., the biophysical properties describing the electrical behavior of the fiber) for a given project is essential. For myelinated fibers, the MRG (McIntyre-Richardson-Grill) model is the gold-standard, and PyFibers includes three MRG variants. The "MRG-discrete" option defines ultrastructure dimensions for a list of specific fiber diameters (5.7 to 16 μm) in the original published model [17], and later extrapolations to 2 μm [27] and 1 μm [11] diameters. The "MRG-interpolation" option defines linear and quadratic fits to the ultrastructure dimensions of the MRG-discrete model to enable simulation of arbitrary fiber diameters between 2 and 16 μm [16]. The smallest diameter in the MRG-discrete model based on experimental data (rather than extrapolations) was 5.7 μm; however, the majority of fibers in peripheral nerves are 1 to 6 μm. Therefore, the Peña model defines ultrastructural dimensions based on experimental data for small diameter, thinly myelinated fibers from 1.011 to 16 μm [18]. PyFibers also includes the Sweeney model, a foundational example for computational modeling of mammalian myelinated fibers [19].

For unmyelinated fibers, PyFibers includes 7 published models. A detailed study [10] compared 5 of these models: Rattay based on non-mammalian experimental data [20], Tigerholm to model cutaneous afferents [23], Sundt to model stimulation of the dorsal root ganglion [24], and multicompartment versions of two Schild variants to model vagal afferents [21,22]. The authors found that, among these 5 models, the Tigerholm model best matched experimental action potential

**Table 2. Myelinated and unmyelinated fiber models available in PyFibers.**

| | Fiber model | $n_{sec}$ per node | Allowed [recommended] diameter (μm) | Description | Described in | Code adapted from |
|---|---|---|---|---|---|---|
| Myelinated | MRG-discrete | 11 | 1, 2, 5.7, 7.3, 8.7, 10, 11.5, 12.8, 14, 15, 16 [≥5.7] | Mammalian fibers | [17] | [16] |
| | MRG-interpolation | 11 | 2-16 [≥5.7] | Mammalian fibers | [17,16] | [16] |
| | Peña | 11 | 1.011–16 [<5.7] | Small diameter mammalian fibers | [17,18] | [16,18] |
| | Sweeney | 2 | Any [10] | Mammalian fibers | [19] | Original to this work |
| Unmyelinated | Rattay | 1 | Any [0.5–2] | Squid giant fiber, adjusted to 37°C | [20] | [10] |
| | Schild 1994 | 1 | Any [0.5–2] | Vagal afferents | [10,21] | [10] |
| | Schild 1997 | 1 | Any [0.5–2] | Vagal afferents | [10,21,22] | [10] |
| | Tigerholm | 1 | Any [0.5–2] | Cutaneous afferents | [23] | [10] |
| | Sundt | 1 | Any [0.52] | Cutaneous afferents | [24] | [10] |
| | Thio cutaneous | 1 | Any [0.5–2] | Cutaneous C-fibers | [25] | [25] |
| | Thio autonomic | 1 | Any [0.5–2] | Autonomic C-fibers | [25] | [25] |

For each model, the table details the number of sections per node ($n_{sec}$ per node, see Fig 3), the allowable and recommended fiber diameters, a brief description of the model, the original publications where the model were described, and the sources from which the model code was adapted.

duration, action potential shape, and strength-duration data, but failed to replicate experimental recovery cycle data. PyFibers also includes models of cutaneous and autonomic unmyelinated fibers from a recent publication that addressed the shortcomings of these prior models by replicating experimental conduction velocity, chronaxie of the strength-duration curve, action potential duration, refractory period, intracellular threshold, and recovery cycle [25]. For all included unmyelinated fiber models, users can choose any diameter, but values larger than 2 μm trigger a warning given the expected range of mammalian unmyelinated fiber diameters [28–31].

**Defining a model fiber:** The `build_fiber()` function in `fiber.py` creates a model fiber as a non-branching series of connected NEURON sections, parameterized according to the specified fiber model. In PyFibers, a "node" refers to one of these serially connected sections that is meant to represent a portion of the fiber that would be excitable (e.g., nodes of Ranvier). Fibers are either "homogeneous" wherein all sections have the same properties or "heterogeneous" wherein different sections may have different properties. Thus, for homogeneous (typically unmyelinated) fibers, nodes and sections are synonymous, and for heterogeneous (typically myelinated) fibers, nodes refer to sections defining nodes of Ranvier between myelinated portions of the fiber (Fig 3).

Spatial discretization of the model fiber can affect simulation accuracy. Section lengths in heterogeneous fibers are specified by the fiber model. For homogeneous fibers, users can provide the section length as an argument to `build_fiber()` (default of 8.333 μm as in [10]), which should be selected such that simulation results are not altered by using shorter sections (S1 Fig). In the same vein, fiber length should be chosen such that increasing length does not change simulation results (S1 Fig). Additionally, the sealed end condition imposed by constructing fiber models of finite length artifactually increases the excitability of nodes at the end of a model fiber. This can lead to non-biophysical "end excitation" of these terminal nodes (S1 Text). When creating a fiber, the user can specify an integer number of passive end nodes to reduce end excitation; the requested number of passive nodes (default 1) on each end of the fiber are stripped of their nonlinear mechanisms and assigned linear, non-excitable mechanisms (described in our documentation (https://wmglab-duke.github.io/pyfibers/fiber_models.html#passive-end-nodes)).

**Recording state variables and adding intrinsic activity:** PyFibers enables recording of various fiber responses to stimulation. The number of action potentials detected at each node using NEURON APCount objects is always recorded,

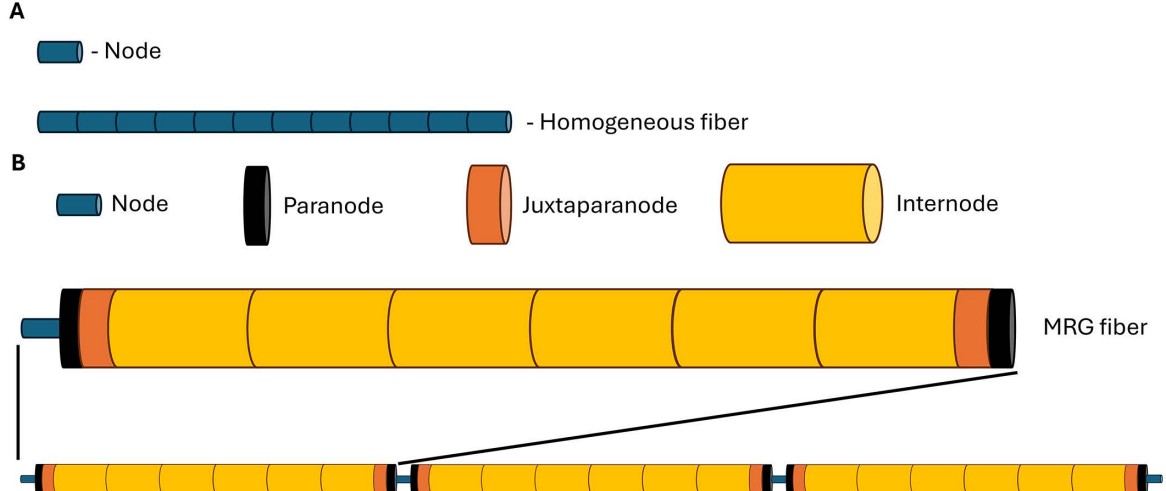

**Fig 3. Construction of a model fiber in PyFibers.** A) For homogeneous fibers (e.g., unmyelinated fibers), the fiber model includes a single set of mechanisms and ultrastructure that is repeated for the instructed number of sections. B) Fibers with multiple types of sections (e.g., MRG as shown) have a repeating series of sections with heterogeneous membrane mechanisms and ultrastructure, defined in the model as a sequence from one node until just before the next node. This sequence is repeated to one less than the instructed number of nodes, and a node is added to the end of the fiber for symmetry. Example shown: MRG model fiber with 4 nodes and 11 sections per node, resulting in a final section count of $n_{sections} = (n_{nodes} - 1)*11 + 1 = (4 - 1) * 11 + 1 = 34$. Figure is not to scale.

and users can instruct recording of state variables, including transmembrane voltage (`Fiber.record_vm()`), gating parameters (`Fiber.record_gating()`), transmembrane current (`Fiber.record_im()`), extracellular voltage (`Fiber.record_vext()`), or any other section properties (`Fiber.record_variables()`) using NEURON vectors. For each function, users can specify the recording locations (nodes only (default), all sections, or specific sections) and times (at every simulation time step, a larger time step than used for simulation, or at specific times).

The `Fiber.add_intrinsic_activity()` method can add a NEURON synapse object that evokes activity in a node at regular intervals or as a Poisson process. Adding intrinsic action potentials to a fiber simulation enables study of the interaction of ongoing activity with extracellular stimulation. For example, testing how ongoing fiber activity affects the excitability of a fiber, or testing for conduction block by delivering intrinsic activity at one end of the fiber and monitoring for action potential propagation at the other end of the fiber (Box 3 and Fig 6). To reduce potential interaction between the stimulus used to generate the intrinsic activity and other extracellular stimuli, `Fiber.add_intrinsic_activity()` evokes action potentials by altering the local membrane conductance (using NEURON's synapse "ExpSyn" mechanism) rather than by injecting an intracellular current (i.e., the method that the `IntraStim` class uses for intracellular stimulation).

**stimulation.py: Executing simulations of electrical stimulation of model fibers.** The `stimulation.py` module enables the user to create a set of instructions for applying intracellular or extracellular electrical stimulation to a model fiber and for calculating the fiber's response (see the module's documentation at https://wmglab-duke.github.io/pyfibers/autodoc/stimulation.html). This module provides basic functionality for modeling stimulation as well as executing a threshold search and provides tools for users to write custom simulations.

**Stimulation class: A framework to define a set of instructions for applying electrical stimulation to model fibers:** The `Stimulation` class provides functionality for initializing and running a simulation, as well as executing a bisection search for activation and block thresholds. `Stimulation` has two primary user-facing methods: `run_sim()` and `find_threshold()`. `Stimulation.run_sim()` simulates the fiber response to stimulation over time. In the Stimulation class, `run_sim()` is merely a placeholder and must be defined by an inheriting subclass (such as the `ScaledStim` class and `IntraStim` class described below) or by a user's custom simulation paradigm (see "Developing custom simulation code to leverage PyFibers model fibers"). `Stimulation.find_threshold()` repeatedly calls `run_sim()` in a bisection search to determine the minimum stimulation amplitude at which either activation or block occurs.

When instantiating `Stimulation` or one of its subclasses, the user must provide a simulation time step and end time. A sufficiently short time step is required for accurate simulations (S1 Fig); however, a time step that is too small can require excess compute resources without improving accuracy. Therefore, it is prudent to determine the longest acceptable time step, considering the time course of the stimuli and the dynamics of the neural responses. The stimulation end time must be sufficiently long to detect responses of interest (e.g., to enable action potentials to propagate from the point of initiation to the recording location). When allowing fiber models to reach steady state (i.e., before applying stimulation), a large time step can be used; `Stimulation` defaults to using a time step of 5 ms from t = −200–0 ms to ensure steady state at t = 0.

After instantiation, `Stimulation.find_threshold()` allows the user to calculate the stimulation threshold for activation or conduction block, executed in two phases: a bounds search and a bisection search (Figs 4 and S2, see also our documentation: https://wmglab-duke.github.io/pyfibers/algorithms.html). The user provides initial upper and lower bounds, ideally bracketing the threshold and skipping the bounds search phase. Otherwise, `find_threshold()` searches upwards if both bounds are subthreshold and downwards if both bounds are suprathreshold. Once the bounds straddle the threshold, a bisection search is executed, which returns the upper bound once the bounds are within a user-specified tolerance (default 1% difference). By default, `find_threshold()` monitors for action potentials at the node closest to 90% fiber length; it is good practice to set this detection location far away from the stimulus to avoid mistaking ohmic changes in transmembrane potential as an action potential.

For activation, threshold is often determined for a single stimulus, requiring ≥1 action potential to consider a stimulus suprathreshold. Users can specify a greater number of action potentials (`num_thresh_aps`) that must be detected for

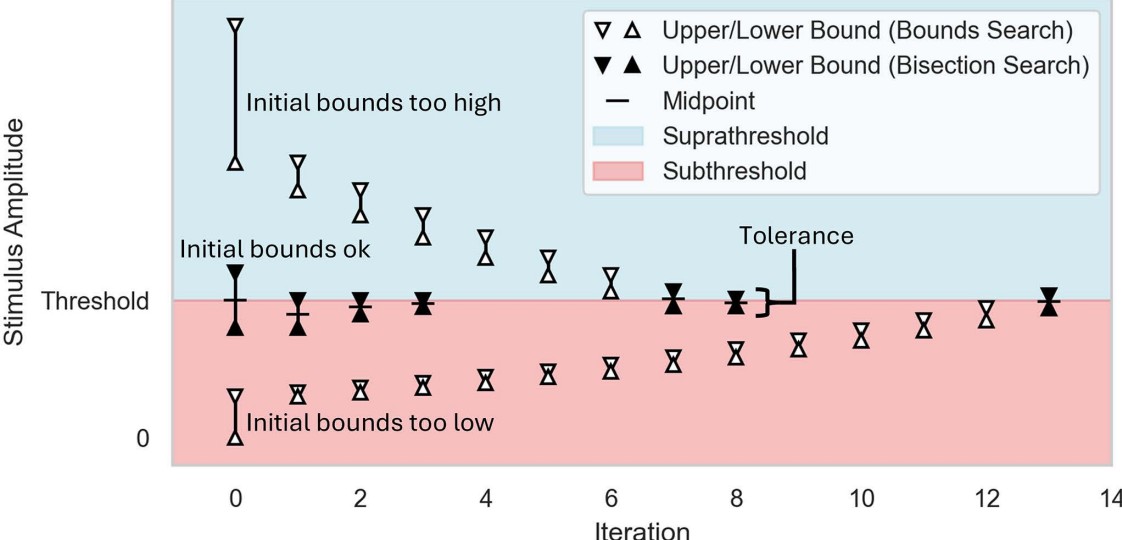

**Fig 4. Three example threshold searches superimposed on the same axes—one with both initial bounds too low, one with both too high, and one with initial bounds straddling the threshold.** For the cases with the initial bounds both too low or too high, a bounds search (white triangles) is first conducted to identify bounds that straddle the threshold, and then the bisection search (black triangles) is initiated. In the case where the initial bounds straddle the threshold, a bisection search begins immediately. The bisection search continues until the difference between the upper and lower bounds is less than the specified search tolerance, and threshold is then considered as the upper bound amplitude.

suprathreshold stimuli. For example, when measuring the recovery cycle, two stimulation pulses are delivered and thus two action potentials are required for suprathreshold stimuli.

Searches for block threshold require more care than searches for activation threshold. In addition to the extracellular blocking signal, intrinsic activity must be evoked at one end of the fiber ("test stimuli") to determine whether it propagates to the other end or is blocked (Fig 6 and Box 3). The initial upper bound of the block threshold search must not be too high to avoid the "re-excitation" regime; kilohertz frequencies can *evoke* activity rather than blocking activity at amplitudes both lower and higher than amplitudes that generate block [11,32]. Action potentials are also typically evoked at the onset of a kilohertz frequency signal, before entering a block regime (Fig 6); therefore, block of the intrinsic activity must be verified after the onset response (via the `block_delay` argument). The built-in bounds search (S2 Fig) assumes a given amplitude is below block threshold if it evokes action potentials after the `block_delay`.

Note that instances of the `Fiber` and `Stimulation` classes are designed to be implemented independent of one another. An instance of the `Stimulation` class defines a set of instructions for applying stimulation to a model fiber. In PyFibers, `Stimulation.run_sim()` and `Stimulation.find_threshold()` operate on a fiber instance, thereby enabling users to test many stimulation paradigms (i.e., many instances of the `Stimulation` class) with a single model fiber. Conversely, one instance of the `Stimulation` class (i.e., instructions for stimulation) can be applied to many model fibers.

**ScaledStim class: Defining extracellular stimulation of model nerve fibers:** The `ScaledStim` class enables extracellular stimulation of model fibers. ScaledStim calculates extracellular potentials using 3 components: (1) spatial distribution of extracellular potentials applied to a model fiber (see "Calculating extracellular potentials for fiber modeling"), (2) stimulation waveform(s) provided to the `ScaledStim` class at instantiation, and (3) a specific stimulation amplitude, which serves as a scaling factor for the spatiotemporal combination of (1) and (2) (Fig 5). Users must carefully consider how the stimulation polarity is defined across these components to ensure that the intended stimulation is delivered to the fiber. Waveforms are provided by the user as `Callables` (e.g., functions) that accept the time in milliseconds as input and return the waveform value at that time (by convention, with a maximum absolute value of 1).

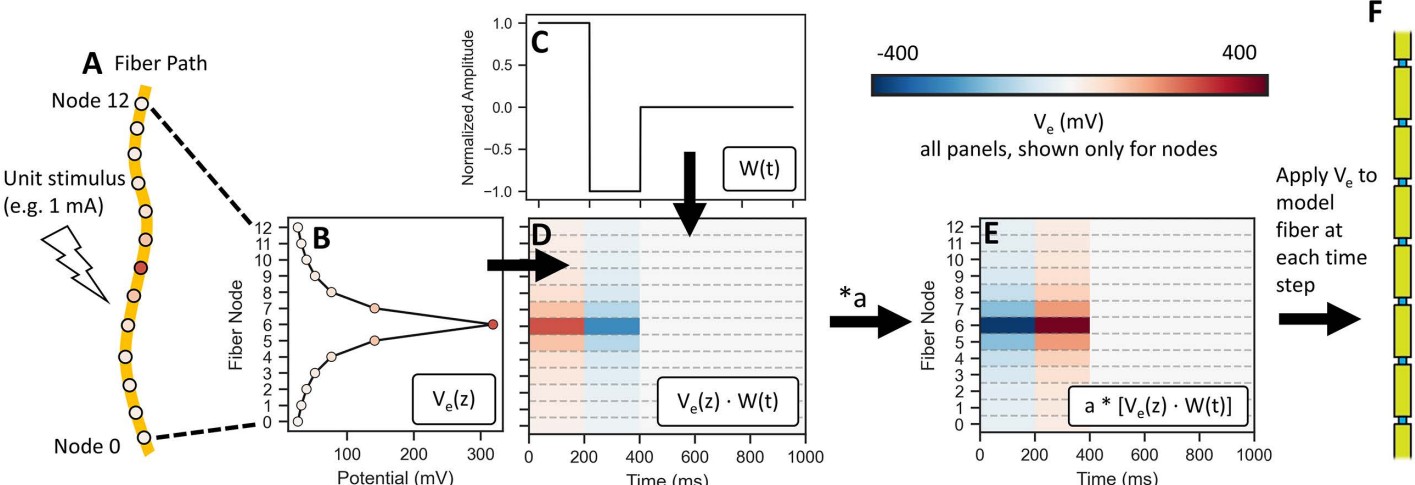

**Fig 5. Process of calculating the spatiotemporal profile of extracellular potentials applied to the model fiber by ScaledStim.run_sim(), which incorporates the spatial distribution of potentials in response to 1 mA stimulation ($V_e(z)$), the unit waveform ($W(t)$), and the stimulation amplitude (a).** If potentials from multiple sources and the corresponding waveforms are provided, steps A-D are performed for each combination of source/waveform/stimulation amplitude, and the results are summed across sources before proceeding. A) Path for non-branching fiber with an arbitrary trajectory in 3D space, with potentials (given by the colored dots) in space generated for a unit stimulus. B) Electrical potential at each section of the fiber. C) Unitless stimulation waveform, by convention defined with a maximum magnitude of 1. D) Matrix multiplication of spatial distribution of potentials (dim: $n_{sections}$ x 1) and waveform (sampled at every timestep to an array of: 1 x $n_{timesteps}$) to obtain $V_e(z,t)$ for a unit stimulus (e.g., 1 mA). E) $V_e(z,t)$ scaled by (signed) amplitude "a". In this example, a=−1.5, so the final stimulus amplitude of the first phase of the symmetric biphasic pulse is −1.5 mA. F) $V_e(z,t)$ applied to a cable model of a myelinated fiber.

`ScaledStim.run_sim(amplitude(s), fiber)` simulates the fiber's response to an extracellular stimulus by multiplying the extracellular potentials by the user-defined waveforms and stimulation amplitude (i.e., scaling factor of the waveform and input potentials), and applying the resulting potentials to the fiber at each timestep (Fig 5 and S2 Text). This assumes quasi-static conditions [33,34] and purely ohmic properties of the tissue around the fiber [35]. Note that the `Fiber` class instances on which `ScaledStim` operates do not require the quasistatic assumption, and users may use the `Fiber` class in more complex simulations by creating their own simulation code for applying the relevant extracellular potentials over time (see "Developing custom simulation code to leverage PyFibers model fibers").

If the model fiber has multiple sets of potentials (i.e., from different sources; S2 Text), the user must provide a matching number of waveforms, and may provide either a single stimulation amplitude that will be applied to all sources or a list of stimulation amplitudes containing one value for each source. In this case, `find_threshold()` only supports searching for the threshold of a single stimulation amplitude; thus, the amplitude for each call of `run_sim()` in a threshold search is applied uniformly to all sources.

After completing the simulation, `run_sim()` returns the number of action potentials detected at a specific node and the time of the last action potential detected; by default, action potential detection is defined as a transmembrane voltage crossing −30 mV with a rising edge at the node closest to 90% fiber length.

**IntraStim class: Defining intracellular stimulation of model nerve fibers:** The `IntraStim` class enables intracellular stimulation by injecting current into a selected section of a model fiber. In the present implementation, `IntraStim` is limited to repeating square current injections; this is distinct from intrinsic activity added to a model fiber via `Fiber.add_intrinsic_activity()`. Instead of a waveform, the user provides keyword arguments to define the stimulation parameters, including location on the fiber, pulse repetition frequency, and pulse duration. Using `IntraStim.find_threshold()` or `IntraStim.run_sim()` will scale the intracellular stimulus amplitude. In the present implementation, `IntraStim` and `ScaledStim` cannot be used in concert; these classes provide a starting point for users wishing to implement more complex stimulation paradigms.

**Calculating extracellular potentials for fiber modeling.** PyFibers requires the extracellular potential at the center of each fiber section to simulate responses to extracellular stimulation and single fiber action potentials from recording electrodes. Users can calculate these potentials analytically using PyFibers or using an external method (e.g., a finite element model). We summarize the process briefly in this section and in more detail in our documentation (https://wmglab-duke.github.io/pyfibers/extracellular_potentials.html).

PyFibers supports the creation of a fiber as 1D (via `build_fiber()` with an input length) or 3D (via `build_fiber_3d()` with an input array of (x,y,z) coordinates describing the path of the fiber through space). Regardless of the fiber dimensionality, NEURON internally simulates a one-dimensional cable. Therefore, both 1D and 3D fibers have a set of coordinates describing the fiber's location in space (`Fiber.coordinates`) and a set of coordinates for the locations of the center of each section along the fiber (`Fiber.longitudinal_coordinates`). By default, 1D fibers are created at $x=0$, $y=0$, and extend from $z=0$ in the positive $z$ direction; `Fiber.set_xyz()` can be called to alter the (x,y) coordinates of a 1D fiber from the default of 0 and to shift the fiber along the $z$ (longitudinal) axis.

Once the fiber geometry is established, users can calculate or load extracellular potentials. Electrical potentials are applied to a fiber via the `Fiber.potentials` attribute and should be generated by a unit stimulus (e.g., 1 mA). For stimulation from a single source, the user provides a 1D array of electrical potentials, one value for each section of the fiber. For stimulation from multiple sources, if each source uses the same waveform (including sources with varying polarities or amplitudes of the same waveform), then the user can weight the potentials in advance under the assumption of linearity; as before, the potentials are then applied as a 1D array. For stimulation where each source delivers a different waveform, the user can provide a 2D array of spatial potential distributions, with dimensions $n_{sources}$ x $n_{sections}$.

To obtain potentials analytically, `Fiber.point_source_potentials()` calculates extracellular potentials ($V_e$) at each section of the model fiber from a point current source, assuming an infinite, homogeneous medium with either isotropic (Equation (1)) or anisotropic (Equation (2)) conductivity:

$$V_e(r) = \frac{I}{4\pi\sigma r}$$

(1)

where $I$ is the current amplitude in mA, $\sigma$ is the isotropic conductivity of the medium in S/m, and $r$ is the distance between the point current and the center of a given section of the model fiber in μm.

$$V_e(x, y, z) = \frac{I}{4\pi\sqrt{\sigma_y\sigma_z x^2 + \sigma_x\sigma_z y^2 + \sigma_x\sigma_y z^2}}$$

(2)

where the anisotropic conductivity of the medium is given by $\sigma_x$, $\sigma_y$, and $\sigma_z$, and the distance from the point source to the center of a given section of the model fiber (in μm) is given by x, y and z.

If users obtain potentials from an external method, such as a finite element model, then the potentials should be sampled at the center of each section using the arc lengths along the fiber's path (from `Fiber.longitudinal_coordinates`) or the (x,y,z) coordinates in 3D space (from `Fiber.coordinates`). It may be undesirable to sample repeatedly electrical potentials along a given fiber path for each newly tested fiber diameter, fiber model, or longitudinal alignment. Further, users may wish to sample potentials along a fiber path in advance, without foreknowledge of the specific required locations of fiber coordinates. Thus, a user may instead sample potentials along the fiber path at high spatial resolution (e.g., 5 μm spacing; S1 Fig) and then interpolate the potentials at the center of each fiber section as needed. The `resample_potentials()` method performs interpolation using the arc-coordinates (1D coordinates along the fiber path) of each potential along the fiber trajectory. For 3D fiber trajectories, the user may instead use `resample_potentials_3d()` which requires a list of potentials and array of 3D coordinates along the fiber path. Users should verify that the sampled potentials have sufficient spatial resolution such that their results (e.g., threshold current) do not change when potentials are obtained at a higher resolution along the fiber (S1 Fig).

**Modeling recording of action potentials.** PyFibers also enables modeling of single fiber action potentials (SFAPs), i.e., the recording of extracellular potentials resulting from a propagating action potential, using methods described in [18], shared in [36], and demonstrated in our documentation tutorial "Recording single fiber action potentials" and in this publication under "Leveraging PyFibers with ASCENT to simulate activation and recording of a model nerve". Briefly, users must enable recording of transmembrane current for all fiber sections (`Fiber.record_im(allsec=True)`) and recording of extracellular potentials (`Fiber.record_vext()`), and then call the `record_sfap()` method, which calculates the corresponding time series signal (in μV) generated at the recording electrode from fiber activity; this method requires extracellular potentials corresponding to a 1 mA stimulus from the recording electrode, which can be computed analytically using `Fiber.point_source_potentials()` or from an external source (see "Calculating extracellular potentials for fiber modeling").

**Tools for analysis after simulation.** PyFibers includes several tools for accessing and analyzing simulation data. Users can access recorded data through the relevant fiber attributes (e.g., `vm=Fiber.vm`). Users should be cautious: since fiber data—gating variables, membrane voltage, action potential counters—are stored in the `Fiber` class instance using NEURON vectors, they are cleared each time `run_sim()` is called. To keep these data after a simulation, data must be copied (e.g., `vm=np.array(Fiber.vm)`) or saved to a file. The user can plot these time series data and conduct additional analyses. For example, `Fiber.measure_cv(start_node, end_node)` calculates action potential conduction speed along the fiber.

**User extension of functionality provided in PyFibers. Developing custom fiber models for use in PyFibers:** PyFibers enables specification of new fiber models with little code, and thus users can implement and use their novel fiber models, as detailed in our documentation (https://wmglab-duke.github.io/pyfibers/custom_fiber.html). An example of how to implement a custom fiber model is provided in the section "Implementing new fiber models". Briefly, users create a new fiber model as a subclass that inherits from the `Fiber` class, including a method for each section type comprising the model (e.g., `MyFiberClass.create_node()`, `MyFiberClass.create_myelin()`); the user then specifies the order in which these functions should be used to create fiber sections (Fig 3). To publish a fiber model for public consumption, users should make new fiber models available as a plugin for PyFibers (also documented on the custom fiber models page referenced above) as a plugin, the new fiber model is contained in its own code repository and is automatically available in PyFibers upon installation. Finally, since NEURON is the foundation of PyFibers, users can define more complex neuronal ultrastructure with NEURON. For example, fibers may be connected to form networks, or users may modify sections and their associated variables; users may need to implement custom simulation code to accommodate these structures.

**Developing custom simulation code to leverage PyFibers model fibers:** We structured PyFibers to enable users to script custom simulations. For example, users wishing to avoid the assumptions of the quasistatic approach [34] may opt to develop their own simulation code to leverage the Helmholtz equation [35]. There are several ways to write custom simulation code for PyFibers: (1) Users can provide a custom simulation function to the `Stimulation` class at instantiation; calling `Stimulation.run_sim()` will then call the custom function. (2) Users can define a subclass inheriting from Stimulation and override `run_sim()` with their own method in the new subclass (for example, `ScaledStim.run_sim()` and `IntraStim.run_sim()` override the default `run_sim()` method of the `Stimulation` class). (3) Users can develop their own simulation paradigms using NEURON code to operate directly on the sections composing a model fiber (see "Defining a model fiber"). (4) Users can use helper methods from the `Stimulation` class (for tasks such as allowing fiber to reach steady state or post-simulation checks for action potentials) and write their own simulation code to apply extracellular potentials over time. For (1) and (2), the user can still use `find_threshold()` to execute a bisection search using their custom simulation function/class, provided that their custom `run_sim()` function is properly parameterized. For more details, see the documentation on custom simulations (https://wmglab-duke.github.io/pyfibers/custom_stim.html).

## Results

Important characteristics of any open-source modeling software include ease of use, ability to address a variety of problems, and validation of the implementation; we address each of these characteristics in this section. These results establish PyFibers as a robust, reliable platform for modeling peripheral nerve fibers and designing stimulation studies.

## Usability

PyFibers was designed to maximize accessibility while providing an extensive feature set and enabling customization. By migrating legacy NEURON HOC code to Python, PyFibers eliminates the steep learning curve associated with HOC, broadening accessibility to a wider range of users [9]. PyFibers dramatically reduces the amount of code required to create models of peripheral nerve fibers and execute simulations of electrical stimulation; for example, the amount of code needed to script an activation threshold search on a MRG model fiber was reduced by two orders of magnitude (see "Operation"). Using developed libraries like PyFibers streamlines the model development process and minimizes potential coding errors from duplication of efforts [37,38]. Further, PyFibers provides validated Python implementations of many widely used fiber models under a shared framework (Table 2), facilitating the reuse and comparison of models without the need for users to code their own implementations. A new fiber model can be tested in the same simulation code by simply changing the model argument in the fiber creation step. The inclusion of extensive documentation and tutorials ensures that both novice and experienced users can navigate and implement complex simulations. During the development of PyFibers, we incorporated feedback from alpha testers in our group and from beta testers in other research groups. Alpha testers used PyFibers in their research projects and their feedback informed the design of the PyFibers interface and the scope of included features. Beta testers were asked to perform a specific set of tasks (S3 Text), and their feedback improved the clarity of the user interface and thoroughness of the documentation.

## Use cases

Computational models enable analysis and design of neural stimulation, block, and recording approaches. Models can be used to simulate the neural responses to devices and parameters used in existing clinical therapies and preclinical studies [18,39], to examine mechanisms of action of observed phenomena [40,41], and to design electrodes and stimulation parameters to achieve targeted neural responses [42,43]. PyFibers builds on the well-established and widely used NEURON platform to enable easier, faster, and less error-prone implementation of nerve fiber models and simulations to achieve these objectives. In the "Operation" section (Box 2 and Fig 2), we demonstrated an example of using PyFibers to calculate the activation threshold for a model nerve fiber, to analyze the resultant transmembrane voltages, and to generate an SFAP from a point-source recording electrode. In this section, we present additional example use cases of PyFibers, including calculating block thresholds, modifying fiber model sections, using PyFibers with the ASCENT pipeline (including finite element modeling of a nerve and cuff electrode) for stimulation and recording, and implementing a new fiber model.

**Calculating thresholds for kilohertz frequency block.** High-frequency electrical stimulation is an active area of research for its ability to rapidly and reversibly block action potential propagation [12,44–46]. PyFibers enables users to switch easily to calculation of block thresholds rather than activation thresholds. Box 3 demonstrates application of a kilohertz frequency signal to block action potentials, the result of which is shown in Fig 6.

> **Box 3. PyFibers code to simulate the response of a 10 μm diameter MRG myelinated fiber (length of 25 nodes = 27 mm) to a 20 kHz square wave (t = 50 to 100 ms) that is delivered at 2.5 mA, 3 mA, or at multiple amplitudes to identify the threshold amplitude for conduction block. Intrinsic activity is evoked at 10% of the fiber length (t = 15 to 145 ms, every 10 ms) to allow monitoring of propagation to the distal end of the fiber and thus determine whether the kilohertz frequency waveform blocks action potential conduction. The extracellular potentials are from a point current source located halfway along the fiber length at an electrode-fiber distance of 250 μm in an isotropic, homogeneous medium with a conductivity of 10 S/m. Transmembrane potential ($V_m$) recording is enabled before the simulation is run. The ScaledStim instance is used to run simulations**

with the kilohertz signal delivered at amplitudes of 2.5 mA and 3 mA. The bisection search determines the threshold amplitude to block action potential transmission, as detected (by default) at the node closest to 90% fiber length after t = 65 ms; this delay avoids confounds from action potentials that are evoked at the onset of the block signal at t = 50 ms. Before running find_threshold(), the simulation is updated to exit at t = 100 ms, when the block stimulus ends; without this change the algorithm would detect action potentials (registered as subthreshold) after block_delay for every stimulus amplitude. Results are shown in Fig 6.

```python
import scipy
import numpy as np
from pyfibers import build_fiber, FiberModel, ScaledStim

# Build the fiber
fiber = build_fiber(FiberModel.MRG_INTERPOLATION, diameter=10, n_nodes=25)

# Enable recording of transmembrane potential (Vm)
fiber.record_vm()

# Add intrinsic spiking
fiber.add_intrinsic_activity(
    loc=0.1,              # Location along the fiber [fraction of length]
    avg_interval=10,      # Average interval between spikes [ms]
    start_time=15,        # Time to start intrinsic activity [ms]
    num_stims=14,         # Number of intrinsic spikes
)

# Calculate spatial distribution of extracellular potentials from a point current source
fiber.potentials = fiber.point_source_potentials(
    x=0,                      # X-coordinate of the point source [µm]
    y=250,                    # Y-coordinate of the point source [µm]
    z=fiber.length / 2,       # Z-coordinate at mid-length of fiber [µm]
    i0=1,                     # Source current for unit stimulus [mA]
    sigma=10,                 # Conductivity of the medium [S/m]
)

# Define function to generate a 20 kHz square wave (unit waveform amplitude)
def khf_square_wave(t):
    frequency = 20000     # Frequency of stimulus [Hz]
    start = 50            # Start time [ms]
    off = 100             # Off time [ms]
    if t<start or t>off:
        return 0
    else:                 # Otherwise on
        return signal.square(2*np.pi*frequency*t/1000) # /1000 to convert to ms

# Define simulation parameters
time_step = 0.001   # Simulation time step [ms]
time_stop = 150     # Total simulation time [ms]

# Create a ScaledStim instance from the waveform
blockstim = ScaledStim(waveform=khf_square_wave, dt=time_step, tstop=time_stop)

# Define stimulus amplitudes
amplitude_1 = -2.5  # Stimulus amplitude scaling factor [unitless]
amplitude_2 = -3.0  # Stimulus amplitude scaling factor [unitless]
```

```
# Run simulations at two different stimulus amplitudes
# Since the original potentials correspond to 1 mA stimulus,
# amplitude_1=-2.5 corresponds to -2.5 mA, amplitude_2=-3.0 corresponds to -3.0 mA
ap, time = blockstim.run_sim(amplitude_1, fiber)
ap, time = blockstim.run_sim(amplitude_2, fiber)

# Update simulation to stop when block turns off
blockstim.tstop = 100

# Find the threshold stimulus amplitude for conduction block
amp, _ = blockstim.find_threshold(
    fiber,
    condition="block",
    block_delay=65 # Delay before checking for block [ms]
)
```

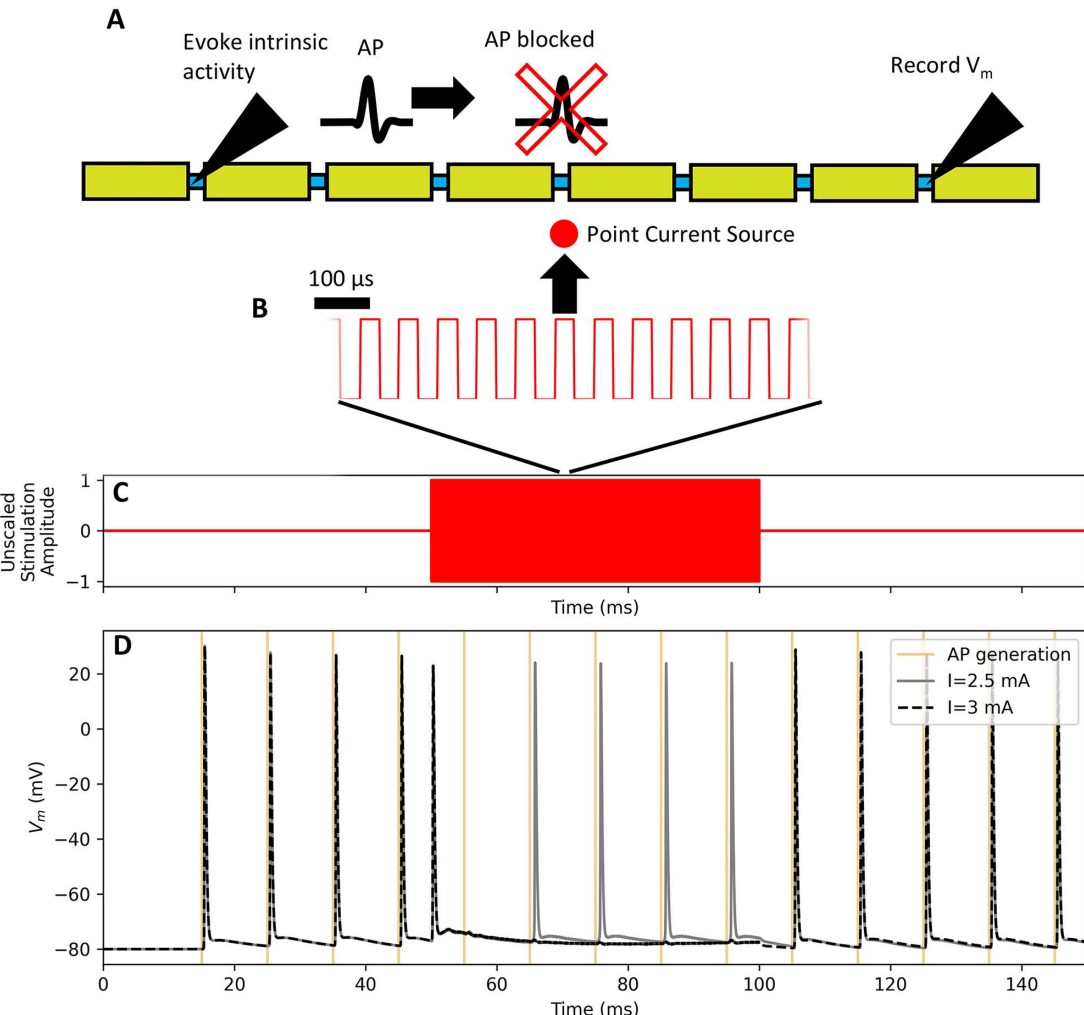

**Fig 6. Simulations of conduction block using the code in Box 3.** AP = action potential; $V_m$ = transmembrane potential. The stimulation and recording setup are shown in panel A; note that the diagram is not to scale and does not show all 25 nodes. A 10 μm MRG fiber with intrinsic activity produced by intracellular stimulation initiated at the node closest to 10% fiber length (from the left), a 20 kHz extracellular square wave (B) delivered via a point source at 50% fiber length at an electrode-fiber distance of 250 μm, and transmembrane potential recorded at the node closest to 90% fiber length. The extracellular signal was delivered from t = 50 to 100 ms (C) at 2.5 and 3 mA and the transmembrane potential at 90% fiber length was recorded (D); only the larger amplitude signal blocked action potential conduction. The search for block threshold in Box 3 yielded −2.81 mA.

**Modifying the properties of the NEURON objects comprising a model fiber.** Users can vary parameters such as ultrastructural dimensions, ion channel conductances, and membrane or myelin capacitances by directly accessing a fiber's NEURON sections; this is distinct from implementing a new fiber model, as described later. As a demonstration, we quantified how changes in ultrastructure affect activation thresholds in the MRG model [16,17] (Box 4 and Fig 7). For this model, the choice of fiber diameter (D) defines the internodal length (INL), number of myelin lamellae (nl), four axonal diameters ($d_{axon\_node}$, $d_{axon\_paranode}$, $d_{axon\_juxtaparanode}$, $d_{axon\_internode}$), and the outer diameter of the myelin sheath ($D_{myelin}$, which is equal to D) (Figs 3 and 7A). In Box 4, we started with a default 10 μm MRG fiber and independently varied: (1) the axonal diameter of the nodes of Ranvier ($d_{axon\_node}$), (2) INL, (3) all diameters ($D_{all}$: $d_{axon\_node}$, $d_{axon\_paranode}$, $d_{axon\_juxtaparanode}$, $d_{axon\_internode}$, and $D_{myelin}$), and (4) all ultrastructural dimensions, as they scale normally with fiber diameter (D). Although internodal length is often considered the primary driver of diameter-dependent excitability, our results show that other ultrastructural dimensions can have comparable or larger effects (Fig 7B).

> **Box 4. PyFibers code to simulate activation thresholds across variations in the ultrastructure of the MRG fiber model (length of 25 nodes). The extracellular potentials were from a point current source located halfway along the fiber at an electrode-fiber distance of 1000 μm in an isotropic, homogeneous medium with a conductivity of 1 S/m. The stimulation waveform was a cathodic 0.2 ms rectangular pulse. We altered the fiber ultrastructure in four ways: (1) $d_{nodes}$: Changing the diameter of the nodes of Ranvier only, (2) INL: Changing the section lengths only, (3) $D_{all}$: Changing the axonal and myelin diameters of all sections. (4) D: Changing the fiber diameter as normal, with its standard effects on all ultrastructural dimensions. Fig 7 shows the effects of these ultrastructural changes on activation thresholds.**

```python
from pyfibers import FiberModel, ScaledStim, build_fiber

# Setup fiber info and create a dictionary of reference fibers
model = FiberModel.MRG_INTERPOLATION
dist = 1000  # Electrode-fiber distance [µm]
diams = [6, 8, 10, 12, 14]  # Fiber diameters (D) [µm]
reference_fibers = {  # Fibers to reference for ultrastructure values
    diam: build_fiber(diameter=diam, fiber_model=model, n_nodes=25) for diam in diams
}

# Stimulation parameters
waveform = lambda t: 1 if t <= 0.2 else 0  # 200 µs pulse
dt, tstop = 0.005, 10  # [ms]
stim = ScaledStim(waveform=waveform, dt=dt, tstop=tstop)

def run_threshold(f, ref_f=None):  # Helper function to run threshold search
    if ref_f is not None:  # If a reference fiber is provided, use its coordinates
        f.potentials = ref_f.point_source_potentials(0, dist, ref_f.length / 2, 1, 1)
    else:  # Otherwise, use the fiber's own coordinates
        f.potentials = f.point_source_potentials(0, dist, f.length / 2, 1, 1)
    return stim.find_threshold(f)[0]
```

```
# Initialize lists to store threshold values
threshes_d_node, threshes_inl, threshes_d_all, threshes_normal = [], [], [], []

for diam in diams:
# SECTION 1: Vary the axonal diameter at the nodes of Ranvier
    # Build a new fiber each time to ensure a clean starting point
    fiber = build_fiber(diameter=10, fiber_model=model, n_nodes=25)
    for refsec, sec in zip(reference_fibers[diam].sections, fiber.sections):
        if sec in fiber.nodes:  # Only change diameters at nodes
            sec.diam = refsec.diam  # Change diameter to reference diameter
    threshes_d_node.append(run_threshold(fiber))

# SECTION 2: Vary internodal length (INL)
    fiber = build_fiber(diameter=10, fiber_model=model, n_nodes=25)
    for refsec, sec in zip(reference_fibers[diam].sections, fiber.sections):
        sec.L = refsec.L  # Change length to reference length
    threshes_inl.append(run_threshold(fiber, ref_f=reference_fibers[diam]))

# SECTION 3: Vary all diameters
    fiber = build_fiber(diameter=10, fiber_model=model, n_nodes=25)
    for refsec, sec in zip(reference_fibers[diam].sections, fiber.sections):
        sec.diam = refsec.diam  # Change diameter to reference diameter
    threshes_d_all.append(run_threshold(fiber))

# SECTION 4: Vary fiber diameter (D) with standard ultrastructure changes dimensions
    fiber = build_fiber(diameter=diam, fiber_model=model, n_nodes=25)
    threshes_normal.append(run_threshold(fiber))
```

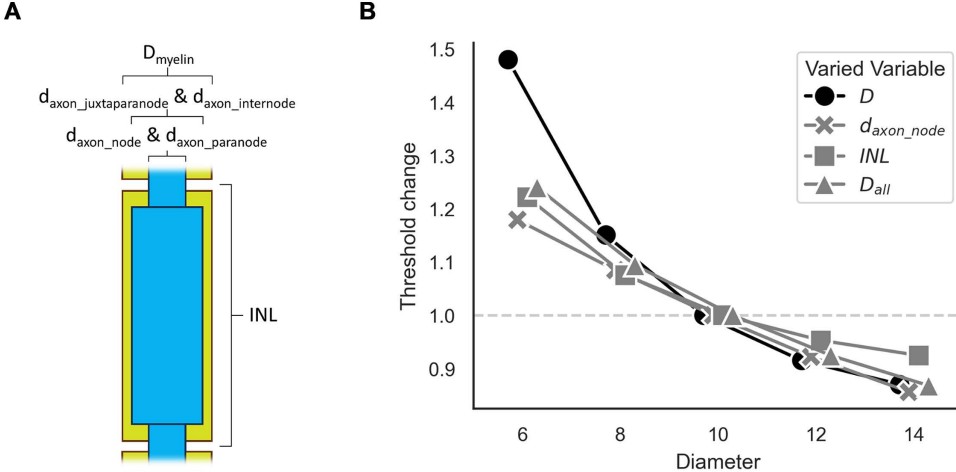

**Fig 7. Effects of modifying fiber ultrastructure on activation thresholds of an MRG fiber, using the code in Box 4. A)** Diagram of ultrastructure parameters modified during the analysis. Note that we did not include an analysis of varying number of myelin lamellae, which is also modified by the fiber diameter (D). B) Effects on activation thresholds normalized by the threshold for an unaltered 10 μm MRG fiber. Parameters not described as changed were held constant. For D, all parameters were varied according to the relationships defined by the MRG-interpolation model. For $d_{axon\_node}$, the diameter of the nodes of Ranvier was varied. For INL, the internodal length (i.e., section lengths) was varied. For $D_{all}$, all axonal and myelin diameters were varied.

**Leveraging PyFibers with ASCENT to simulate activation and recording of a model nerve.** We used PyFibers to replace the HOC fiber simulation code in ASCENT, an open-source pipeline for modeling peripheral nerve stimulation [16]. To exemplify use of ASCENT-PyFibers, we used ASCENT's compound action potential (CAP) tutorial (https://github.com/wmglab-duke/ascent/tree/v1.5.0/examples/cap_tutorial) configuration files without modification (Fig 8). We used ASCENT v1.5.0 [48] with COMSOL v6.1 (COMSOL Inc, Burlington, MA, USA). Briefly, we modeled a 30 cm-long rat vagus nerve instrumented with a MicroLeads two-contact cuff at $z = 4$ cm for stimulation and a MicroLeads three-contact cuff at $z = 15$ cm (the center of the nerve) for recording (Fig 8A). We placed Peña model fibers along the centroid of the nerve across a range of 14 diameters (~1.0–9.8 μm). We delivered stimulation in a bipolar configuration at an amplitude sufficient to activate all fibers and used a monopolar configuration for recording fiber electrical signals. Built-in ASCENT scripts were applied to compute the SFAP for each fiber and a compound nerve action potential (CNAP) across all fibers (Fig 8B and 8C). This example demonstrates the utility of PyFibers within a multi-scale pipeline for modeling neural responses to stimulation. SFAPs and CNAP generated with ASCENT-PyFibers were identical to those produced with ASCENT-HOC (S3 Fig). Further validation of the PyFibers integration into ASCENT is demonstrated in "Validation against previously published nerve fiber model implementations".

The integration of PyFibers will ease the development of new features and fiber models into ASCENT. By relying on PyFibers' well-documented, modular Python framework, developers can more readily implement new simulation protocols and other fiber modeling features. The migration from HOC to Python also improves troubleshooting: Python has interactive debugging capabilities and a clearer structure for identifying and fixing issues, and PyFibers includes extensive unit tests to help with early identification of problems. Overall, these changes dramatically lower the barriers to extending ASCENT's functionality and help ensure the reliability of future releases.

**Implementing new fiber models.** Implementation of new fiber models and stimulation paradigms was designed to require as little user-generated code as possible. In Box 5, we demonstrate how a new model [19] can be implemented with minimal overhead; such a model is then able to leverage the extensive simulation and analysis tools in PyFibers. The basic steps are: create a subclass of `Fiber`, declare model parameters in the initializer, and provide small builder functions that define the section order and geometry (e.g., nodes and myelin) while inserting compiled NEURON mechanisms (assuming that the needed mechanisms have been written in NMODL and compiled prior to runtime). Calling `register_custom_fiber()` exposes the model as an option in the `FiberModel` enum (a registry of available fiber types), making it immediately usable with `build_fiber()` and fully compatible with PyFibers tools for simulation and analysis. This simplicity allows users to focus on model parameterization, rather than having to generate the boilerplate code necessary to implement models. Detailed description of the methodology for building custom fiber models, including specifying them as plugins, can be obtained from our documentation (https://wmglab-duke.github.io/pyfibers/custom_fiber.html).

---

**Box 5. PyFibers code to implement and register a custom myelinated fiber model [19] as a plugin, exposing it as FiberModel.SWEENEY for immediate use with build_fiber and the high-level stimulation/analysis tools. The accompanying validation plots are in S4 Fig and the sweeney.mod mechanism file is S4 Text.**

```python
import neuron
from neuron import h

from pyfibers import FiberModel, ScaledStim, build_fiber, register_custom_fiber
from pyfibers.fiber import Fiber

h.load_file("stdrun.hoc")

# Load the compiled sweeney mechanism
neuron.load_mechanisms("sweeney")
```

```python
class SweeneyFiber(Fiber):
    """Implementation of the Sweeney fiber model."""

    submodels = ["SWEENEY"]

    def __init__(self: SweeneyFiber, diameter: float, **kwargs) -> None:
        """Initialize SweeneyFiber class.

        :param diameter: fiber diameter [microns]
        :param kwargs: keyword arguments to pass to the base class
        """
        assert "delta_z" not in kwargs, "Cannot specify delta_z for Sweeney Fiber"
        super().__init__(diameter=diameter, **kwargs)
        self.gating_variables = {
            "h": "h_sweeney",
            "m": "m_sweeney",
        }
        self.myelinated = True
        self.delta_z = self.diameter * 100
        self.v_rest = -80  # millivolts

    def generate(self: SweeneyFiber, **kwargs) -> Fiber:
        """Build fiber model sections with NEURON.

        :param kwargs: passed to superclass generate method
        :return: Fiber object
        """
        # Function list for section order
        function_list = [
            self.create_node,
            self.create_myelin,
        ]

        return super().generate(function_list, **kwargs)

    def create_node(self: SweeneyFiber, index: int, node_type: str) -> h.Section:
        """Create a node of Ranvier.

        :param index: Section index in the fiber.
        :param node_type: Node type ('active' or 'passive').
        :return: Created node with Sweeney mechanisms
        """
        name = f"{node_type} node {index}"
        node = h.Section(name=name)
        node.nseg = 1
        node.diam = self.diameter * 0.6
        node.L = 1.5  # um
        node.insert("sweeney")
        node.cm = 2.5  # uF/cm^2
        node.Ra = 54.7  # ohm-cm
        node.insert('extracellular')
        node.xc[0] = 0  # short circuit
        node.xg[0] = 1e10  # short circuit
        node.ena = 35.64 # mV
        return node

    def create_myelin(self: SweeneyFiber, index: int) -> h.Section:
        """Create a myelin section.

        :param index: Section index in the fiber.
        :return: Created myelin section
        """
        name = f"myelin {index}"
        section = h.Section(name=name)
        section.nseg = 1
        section.diam = self.diameter * 0.6
        section.L = 100 * self.diameter - 1.5  # um, -1.5 since 100*D is internodal length
        section.cm = 0
        section.Ra = 54.7  # ohm-cm
        section.insert('extracellular')
        section.xc[0] = 0  # short circuit
        section.xg[0] = 1e10  # short circuit
        return section

# Register the custom fiber model with the FiberModel enum
register_custom_fiber(SweeneyFiber)

model = FiberModel.SWEENEY  # type of fiber model

# Create a model fiber
fiber = build_fiber(diameter=10, fiber_model=model, temperature=37, n_nodes=21,
passive_end_nodes=0)
```

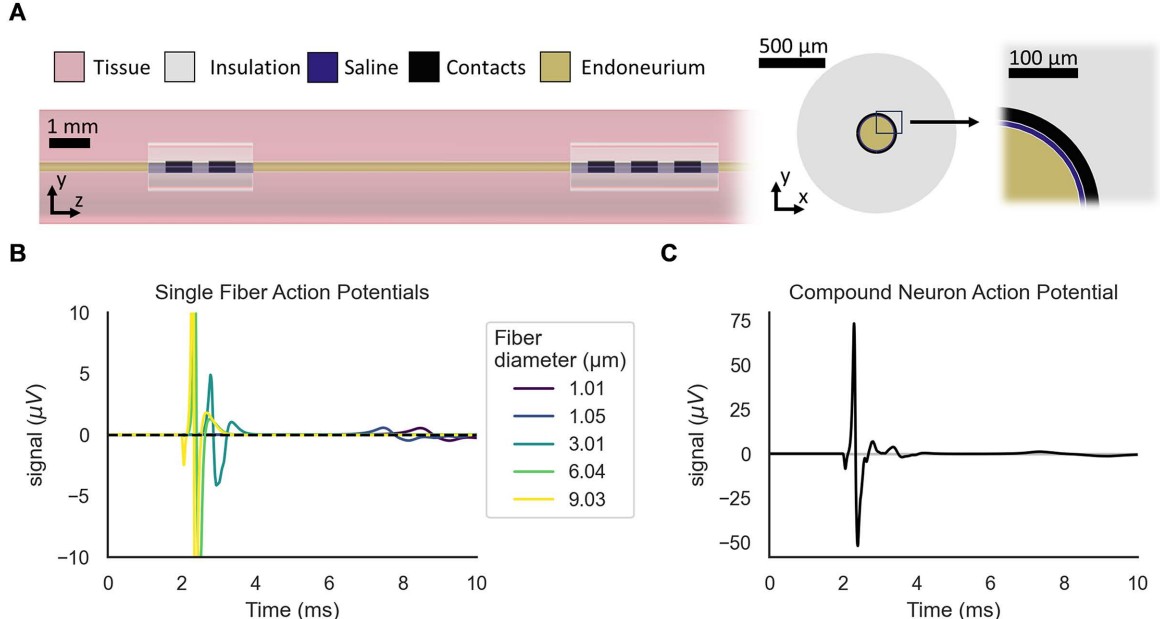

**Fig 8. Modeling of rat vagus nerve stimulation and recording of single fiber action potentials and compound nerve action potentials using the Peña model in ASCENT with PyFibers.** A) Geometrical model instrumented with stimulation (two contacts) and recording (three contacts) cuffs. The cross sectional x-y view does not show the surrounding tissue. The perineurium is modeled using a contact impedance [47] and therefore is not visible as a distinct geometry. B) Single fiber action potentials from select fiber diameters. All fibers were along the nerve centroid. C) Compound action potential resulting from activation of all 14 fibers, calculated as summation of their single fiber action potentials.

## Validation against previously published nerve fiber model implementations

We replicated published responses for each of the 11 fiber models included in PyFibers (Table 2) to ensure the accuracy of our implementations. For all analyses, action potentials were detected when transmembrane voltage crossed −30 mV with a rising edge. To quantify conduction velocity and action potential shape, we stimulated intracellularly at the second node, because we used passive end nodes. To obtain action potential time courses, we recorded the transmembrane voltage at the center node. We calculated conduction velocity as the difference in action potential times at 25% and 75% fiber length divided by the distance between the two nodes. To determine activation thresholds, we checked for action potentials at 90% fiber length and used a bisection search with a tolerance of 1% difference between the upper and lower bounds. For all simulations, we used a time step of 5 μs.

For the MRG fiber models, we compared data from our MRG-discrete and MRG-interpolation implementations to results in the original paper [17], including conduction velocity, afterpotentials, recovery cycle, strength-duration, and current-distance (Fig 9). MRG-discrete replicated the original results precisely, while MRG-interpolation had small differences due to the ultrastructure being defined by polynomial fits. We also validated our implementation of the Peña fiber model, including activation thresholds (Fig 11) and recordings of gating variables and transmembrane potential (S3 Fig). We compared activation responses (S5 Fig) and ultrastructural parameters (S6 Fig) between all three variants of the MRG fiber model.

For five of the unmyelinated fiber models (Tigerholm, Rattay, Sundt, Schild 1994, Schild 1997), we used the methods described in the original paper to simulate conduction speed, action potential shape, strength-duration, and recovery cycle [10], which we replicated with PyFibers (Fig 10). We also verified that our implementations of the Thio autonomic and Thio cutaneous fiber models [25] matched the previously published HOC implementations (S7 Fig), and that our implementation of the Sweeney model [19] matched the original publications' data (S4 Fig).

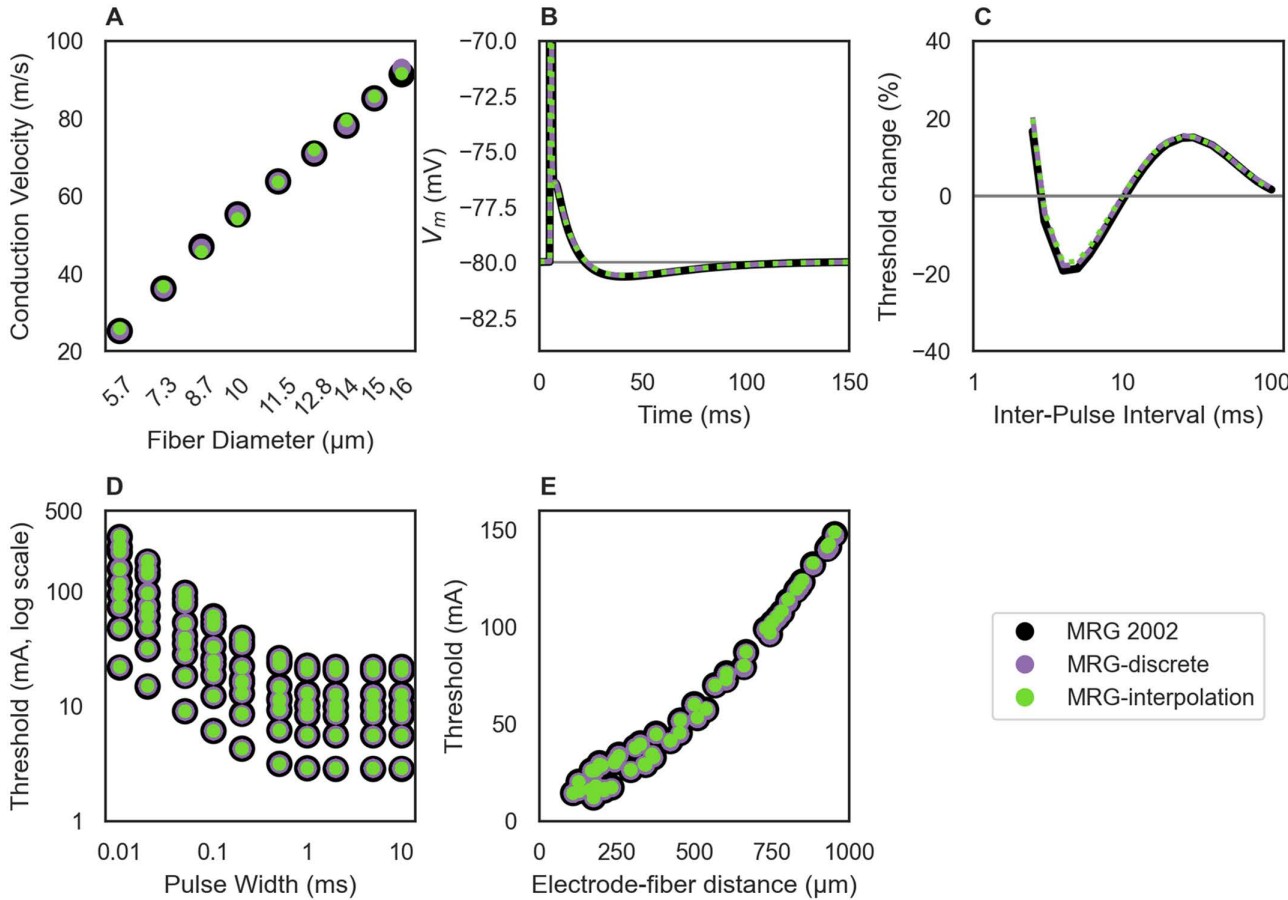

**Fig 9. Replication of published responses for the MRG fiber model.** Data in black were provided by the original authors [17], and data in purple and green are from the MRG-discrete and MRG-interpolation PyFibers implementations, respectively. All fibers were 21 nodes long and except for panel A, all data are for 10 μm diameter fibers. We used an anodic rectangular pulse for intracellular stimulation (panels A-C) and a monopolar rectangular pulse delivered by a cathodic point current source in an anisotropic medium (sigma(x,y,z) = {1/12, 1/12, 1/3} [S/m], where z is along the axis of the fiber (panels D-E). The stimulation pulse began at t = 0 and lasted for 0.1 ms, except for panel C where we evaluated pairs of pulses (1 ms duration each) and panel D where we evaluated different pulse widths. A) Conduction velocity across fiber diameters, calculated using action potential times from fiber(0.25) to fiber(0.75) after intracellular stimulation at fiber[1]. B) Action potential time course, recorded at the center node (fiber(0.5)) after an intracellular stimulus at fiber[1]. C) Recovery cycle for intracellular stimulation delivered at the center node (fiber(0.5)). We first determined activation threshold ($I_{th}$) for a single pulse with 1 ms duration. We then simulated a pair of pulses: one at $t_1 = 1$ ms and $I_{th}$, and a second at $t_2 = 1$ ms + interpulse interval; we determined the activation threshold for the second pulse and its difference relative to $I_{th}$. D) Strength-duration response for electrode-fiber distances from 100 to 500 μm and longitudinal alignment at (1) the center node, (2) shifted by ¼ of the internodal length, (3) shifted by ½ of the internodal length. For the MRG-discrete implementation, we matched the point current source position to the original publication; the z-position was assigned relative to the start of the fiber. The MRG-interpolation implementation has slightly different ultrastructure, including positions of the nodes of Ranvier; therefore, we altered the longitudinal coordinates of the point current sources to maintain the same position with respect to the center node of Ranvier. E) Current-distance response for point sources with electrode-fiber distances from 109 to 953 μm and longitudinal alignment from −563 to +549 μm (with respect to the center node).

We validated the PyFibers integration into ASCENT presented in a previous section ("Leveraging PyFibers with ASCENT to simulate activation and recording of a model nerve") by executing ASCENT's "tutorial" simulation (https://wmglab-duke-ascent.readthedocs.io/en/latest/Getting_Started.html#setting-up-a-run-of-ascent) using ASCENT v1.5.0 [48] with its existing HOC code versus PyFibers. We made the following changes to the "tutorial" simulation: (1) we increased the model length and fiber length from 12.5 mm to 50 mm to avoid end excitation in large-diameter myelinated fibers, (2) we simulated a range of fiber diameters, (3) we simulated all 8 fiber models available in ASCENT, and (4) we decreased

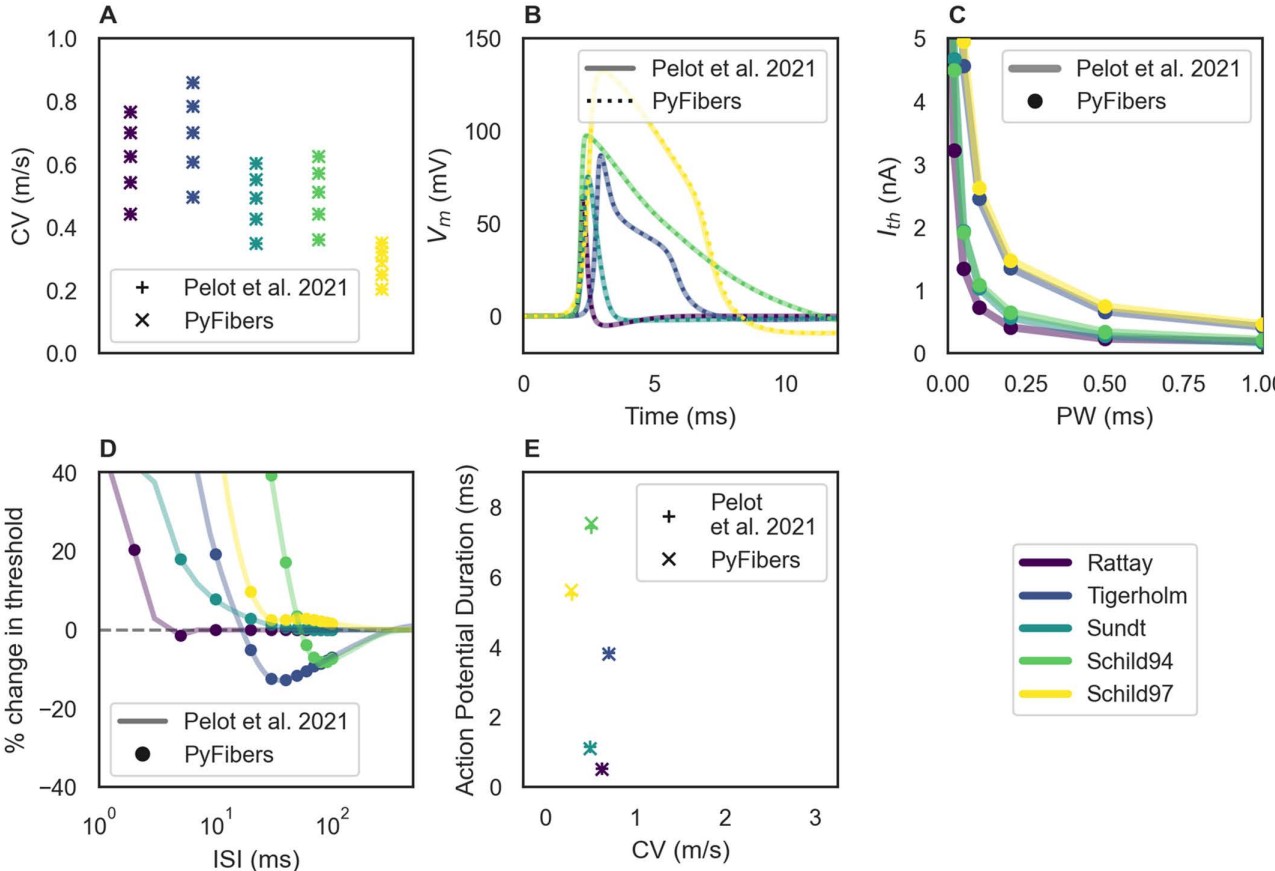

**Fig 10. Replication of published responses for five of the seven unmyelinated fiber models in PyFibers.** Data from [10] were downloaded from the publicly available repository [49]. All fibers were 5 mm long with 600 sections (i.e., each section was 8.33 μm long and comprised a single segment), and except for panel A, all data are for 1 μm diameter fibers. As in [10], we removed all nonlinear mechanisms from the end nodes, except in the Tigerholm model. Stimulation was an intracellular anodic rectangular pulse beginning at t = 1 ms with a pulse width of 0.1 ms, except for panel C where we evaluated different pulse widths and panel D where we evaluated pairs of pulses (0.1 ms duration each). A) Conduction velocity calculated using action potential times from fiber(0.25) to fiber(0.75) after intracellular stimulation at fiber[1]. Each dot represents one fiber diameter (0.5, 0.75, 1, 1.25, 1.5). B) Action potential time course recorded at the center node fiber(0.5) after an intracellular stimulus at fiber[1]. C) Strength-duration curves calculated using intracellular stimulation at fiber(0.5). D) Recovery cycle for intracellular stimulation delivered at the center node (fiber(0.5)). We first determined activation threshold ($I_{th}$) for a single pulse with 0.1 ms duration. We then simulated a pair of 0.1 ms duration pulses: one at $t_1$ = 1 ms and 1.5 * $I_{th}$, and a second 0.1 at $t_2$ = 1 ms + inter-stimulus interval (ISI); we determined the activation threshold for the second pulse and its difference relative to $I_{th}$. E) Action potential duration calculated by taking the earliest and latest times where $V_m$ > baseline $V_m$ + 0.1*(peak $V_m$ − baseline $V_m$).

the tolerance of the bisection search, terminating when the percent difference between the suprathreshold and subthreshold amplitudes was 0.01% instead of 1%.

ASCENT with PyFibers reproduced the HOC-based NEURON activation thresholds (Fig 11A), and all threshold differences were < 0.01%. With PyFibers, the run times for threshold searches were reduced by 58% ± 25% (mean ± SD) across all fiber models (Fig 11B); the underlying performance improvements are detailed in S5 Text. The integration of PyFibers into ASCENT replaced the legacy HOC-based fiber simulation backend and eliminated nearly 8,000 lines of code. We verified that ASCENT with PyFibers reproduced the results from ASCENT with HOC for transmembrane potential and gating variable time course, single fiber action potentials, and compound action potentials (S3 Fig).

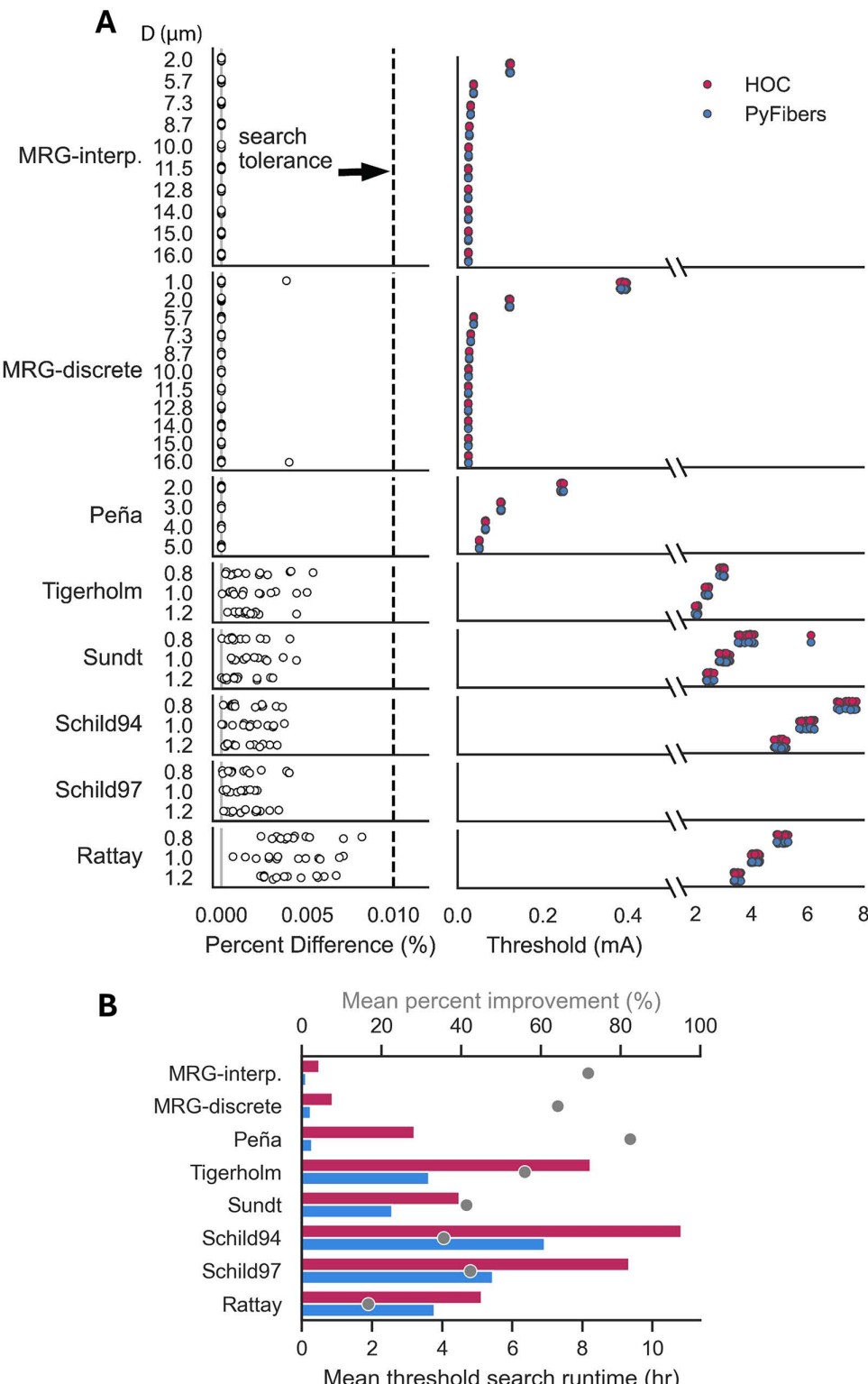

**Fig 11. ASCENT threshold results for the original HOC NEURON versus novel PyFibers implementations.** Adaptation of the ASCENT tutorial task: stimulation of a monofascicular rat cervical vagus nerve with 13 fiber locations, instrumented with a cuff with two circumneural contacts that delivered a biphasic charge-balanced pulse, with a first phase of 100 µs, interphase delay of 100 µs, and second phase of 400 µs. A) Percent differences in thresholds (left) and absolute activation thresholds (right) for each fiber diameter. Percent difference was calculated as the absolute value of the

difference between the HOC and PyFibers thresholds, divided by their mean. B) Threshold search runtimes all executed with the same hardware and compute resources. The bars show the mean runtime across all threshold searches. Dots show the percent improvement of PyFibers over the HOC-based NEURON implementation in ASCENT calculated as 100 * (1 – PyFibers runtime/ HOC runtime).

## Availability and future directions

PyFibers is open-source and publicly available from either PyPI (https://pypi.org/project/pyfibers/) or GitHub (https://github.com/wmglab-duke/pyfibers). PyFibers users should cite both the PyFibers paper and the DOI of the PyFibers code version that was used. Specific versions of the PyFibers code can be cited using Zenodo DOIs from https://doi.org/10.5281/zenodo.17068760. The documentation is hosted on GitHub at https://wmglab-duke.github.io/pyfibers/. The data presented in this publication are available on sparc.science at https://doi.org/10.26275/8ssx-gcil, including the code used to run all simulations, as well as the data and code to generate all figures.

PyFibers was designed to be open source and foster efficient community development and expansion. Open source tools are important for scientific reproducibility and code reusability [50] and can provide important insights into how code can tackle a research problem [51]. Community engagement is key to the continued expansion and maintenance of open-source projects [52]. Availability on PyPI makes installation of PyFibers simple, and hosting the package on GitHub provides a robust platform for community discussion and development. By reimplementing established fiber models and algorithms in a user-friendly package, PyFibers provides an infrastructure for developing and testing simulation protocols. We included the ability for other research groups to publish fiber models as plugins for PyFibers, fostering community development. The ability to be used independently or as part of a larger modeling workflow gives researchers flexibility to adapt PyFibers to their needs.

PyFibers places a strong emphasis on reproducibility and scientific rigor, essential components of any research computing work [38]. The package has comprehensive integration and unit testing to ensure that simulations operate as intended, minimizing the risk of errors introduced by new features or code modifications. By standardizing model implementations and providing clear guidelines for model parameterization, PyFibers enhances the reliability and reproducibility of simulation results. Rigorous validation of fiber models against established benchmarks further underscores accuracy. By successfully replicating published fiber responses, we demonstrated that PyFibers can reproduce the neural dynamics of HOC implementations of multiple fiber models. We ensured transparency and reproducibility by providing publicly available datasets, version-controlled code, and a unique DOI for each forthcoming version of PyFibers [53].

Future versions of PyFibers will focus on providing additional tools to ease the user experience and expanding the available feature set. Providing more tools for post-hoc analysis and plotting will further reduce the coding burden on users. Additional fiber models in the library will allow for broader comparisons and applications in neuromodulation.

In summary, PyFibers offers a cutting-edge, Python-based framework for creating model fibers and simulating the responses of peripheral nerve fibers to electrical stimulation within the NEURON environment. The emphasis on modularity, reproducibility, and ease-of-use, coupled with rigorous validation and a framework for community-driven development, positions PyFibers as a pivotal tool for neural engineering researchers. As neuromodulation therapies continue to evolve, PyFibers will play a crucial role in bridging the gap between computational modeling and practical therapeutic applications, thereby contributing to the development of more effective clinical therapies.

## Supporting information

**S1 Fig. Example of spatial and temporal sensitivity analyses to avoid numerical impacts on modeled thresholds and other output measures.** Effects of spatial and temporal parameters on activation thresholds. Stimulation of a 1 μm diameter Tigerholm fiber with a length of 2 mm. The stimulation potentials were from a point current source located halfway along the fiber at an electrode-fiber distance of 100 μm in an isotropic, homogeneous medium with a conductivity of 1

S/m. The stimulation waveform was a monophasic cathodic rectangular pulse with pulse duration of 1 ms at t = 0. The simulation used a time step of 0.005 ms. Parameters were varied from the preceding description as specified in each panel; for each panel, threshold change was calculated from the "best" parameter. A) Distance between consecutive coordinates of electric potentials from a point current source, which were then resampled to match the distance between the centers of the fiber sections (8.333 μm). B) Fiber length. C) Section length. D) Time step.
(TIF)

**S2 Fig. Flowchart for Stimulation.find_threshold(): Flowchart of PyFibers algorithm to identify activation or block threshold.** For simplification, several validation checks and details of steps are omitted or simplified. Initial upper and lower bound amplitudes are provided as user inputs. If the bounds are too low (both subthreshold), an upwards bounds search commences, and if the bounds are too high (both suprathreshold), a downwards bounds search commences. Once the bounds are established (lower bound subthreshold, upper bound suprathreshold), a bisection search executes until the user-defined exit criterion is reached. Note: During block threshold searches, the sub/suprathreshold check is delayed until after a user provided "block_delay" argument. If the stimulus generates action potentials after this delay, the stimulus will be considered subthreshold.
(TIF)

**S3 Fig. Comparison of fiber responses using ASCENT with HOC or with PyFibers for the Peña fiber model: Comparison between ASCENT with HOC and ASCENT with PyFibers.** A) Comparison of single fiber action potentials and compound action potentials using Peña fibers in ASCENT with HOC (left) and ASCENT with PyFibers (right). Simulation used the ASCENT tutorial files for CAP recording. The bottom row is the sum of SFAPs from the top row. B, C) Comparison of transmembrane potentials (B) and gating variables (C) over time for the center node of a 4 μm diameter Peña fiber in ASCENT with HOC (solid red) and ASCENT with PyFibers (dashed blue), using the model described in "Leveraging PyFibers with ASCENT to simulate activation and recording of a model nerve".
(TIF)

**S4 Fig. Sweeney fiber model validation: Results from simulating the Sweeney fiber model in PyFibers, compared to the data from the original paper.** A) Strength-duration curve. B) Action potential conduction.
(TIF)

**S5 Fig. Activation responses of the three variants of the MRG fiber model: Comparison of 2 μm diameter Peña, MRG-discrete, and MRG-interpolation fibers across conduction velocity, afterpotential shape, recovery cycle, strength-duration curve, and electrode-fiber distance; these data were generated using the same methods as in "Validation against previously published nerve fiber model implementations".**
(TIF)

**S6 Fig. Ultrastructure of three variants of the MRG fiber model: Comparison of ultrastructural parameters across MRG fiber variants, for diameters across the valid range for each model.**
(TIF)

**S7 Fig. Validation of Thio fiber models.** Comparison of Thio fiber model outputs in PyFibers versus published data, confirming that conduction velocity, action potential shape, strength-duration curve, recovery cycle, and action potential duration match previously reported values. We used the same methods as reported in the original publication, including simulating all fibers at 37°C, except for panels C and D, where the Thio cutaneous fiber was simulated at 33°C, and panel E, where action potential duration was measured at 24°C. Except for panel A, all simulations were 1 μm diameter fibers. A) Conduction velocity for 0.5, 1, and 1.5 diameter fibers. B) Action potential shape. C) Strength duration relationship. D) Recovery cycle. E) Conduction velocity vs action potential duration.
(TIF)

**S1 Text. Resolving end excitation.**
(DOCX)

**S2 Text. Scaling of extracellular potentials in ScaledStim.run_sim().**
(DOCX)

**S3 Text. Document with tasks sent to external beta testers.**
(DOCX)

**S4 Text. Sweeney mechanism code.**
(DOCX)

**S5 Text. Bisection search algorithm changes to reduce threshold runtimes.**
(DOCX)

## Acknowledgments

We thank Minhaj Hussain for his advice in exploring NEURON code features. We thank Brandon Thio for valuable discussions related to implementation of his fiber model. We thank Tara Zamani for implementing features supporting single fiber action potential recordings and 3D fiber trajectories. We thank Ari Singh for implementing support for waveforms as Callables. We thank Dr. Cameron McIntyre for providing the original data used in the MRG fiber model publication. We also wish to acknowledge the contributions of our alpha testers—Lucas Kaplan, Newland Zhang, Kathryn Turk, Maria Jantz, Jerry Jiang, Tyler King, and Isha Chugh—whose efforts were instrumental in refining PyFibers. Finally, we wish to acknowledge our beta testers—Sudi Sridhar, Nick Merola, Edgar Peña and his student Luke Ishag, Scott Lempka and his students Adrian Laura and Liam Matthews, and Calvin Eiber—for providing critical feedback that shaped the final release of the package.

## Author contributions

**Conceptualization:** Daniel P. Marshall, Elie S. Farah, Eric D. Musselman, Nicole A. Pelot, Warren M. Grill.

**Data curation:** Daniel P. Marshall.

**Formal analysis:** Daniel P. Marshall.

**Funding acquisition:** Nicole A. Pelot, Warren M. Grill.

**Investigation:** Daniel P. Marshall.

**Methodology:** Daniel P. Marshall, Elie S. Farah, Eric D. Musselman.

**Project administration:** Nicole A. Pelot, Warren M. Grill.

**Resources:** Nicole A. Pelot, Warren M. Grill.

**Software:** Daniel P. Marshall, Elie S. Farah.

**Supervision:** Eric D. Musselman, Nicole A. Pelot, Warren M. Grill.

**Validation:** Daniel P. Marshall, Elie S. Farah.

**Visualization:** Daniel P. Marshall.

**Writing – original draft:** Daniel P. Marshall, Elie S. Farah, Eric D. Musselman.

**Writing – review & editing:** Nicole A. Pelot, Warren M. Grill.

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
