## [Decision Letter · Decision Letter 0]

27 May 2025

PyFibers: An open-source NEURON-Python package to simulate responses of model nerve
fibers to electrical stimulation

PLOS Computational Biology

Dear Dr. Grill,

Thank you for submitting your manuscript to PLOS Computational Biology. After careful
consideration, we feel that it has merit but does not fully meet PLOS Computational
Biology's publication criteria as it currently stands. Therefore, we invite you to
submit a revised version of the manuscript that addresses the points raised during
the review process.

Please submit your revised manuscript within 60 days Jul 27 2025 11:59PM. If you will
need more time than this to complete your revisions, please reply to this message or
contact the journal office at ploscompbiol@plos.org. When you're ready to submit
your revision, log on to https://www.editorialmanager.com/pcompbiol/ and select the
'Submissions Needing Revision' folder to locate your manuscript file.

We look forward to receiving your revised manuscript.

Kind regards,

Alain Nogaret, PhD

Academic Editor

PLOS Computational Biology

Daniele Marinazzo

Section Editor

PLOS Computational Biology

**Additional Editor Comments:**

The software builds upon libraries of the NEURON suite and innovations ought to be
emphasized in the revision. As a "software article" the manuscript could be
published as long as the novelty and examples of impact cases not addressed by
existing libraries could be stated. In your revision please address additional
reviewer comments.

**Journal Requirements:**

3) Your manuscript is missing the following sections: Availability and Future
Directions. Please ensure that your article adheres to the standard Software article
layout and order of Abstract, Introduction, Design and Implementation, Results, and
Availability and Future Directions. For details on what each section should contain,
see our Software article guidelines:

https://journals.plos.org/ploscompbiol/s/submission-guidelines#loc-software-submissions

5) We notice that your supplementary Figures, and Tables are included in the
manuscript file. Please remove them and upload them with the file type 'Supporting
Information'. Please ensure that each Supporting Information file has a legend
listed in the manuscript after the references list.

Potential Copyright Issues:

- The following Figures contain screenshots: Box 1, 2, 3, and 4. We are not permitted
to publish these under our CC-BY 4.0 license, websites are usually intellectual
property and are copyrighted.This includes peripheral graphics of the web browser
such as icons and button. We ask that you please remove or replace it.

7) Please ensure that the funders and grant numbers match between the Financial
Disclosure field and the Funding Information tab in your submission form. Note that
the funders must be provided in the same order in both places as well. State the
initials, alongside each funding source, of each author to receive each grant. For
example: "This work was supported by the National Institutes of Health (####### to
AM; ###### to CJ) and the National Science Foundation (###### to AM).".

**Reviewers' comments:**

Reviewer's Responses to Questions

**Comments to the Authors:**

Reviewer #1: The submission “PyFibers: An Open-Source NEURON-Python Package to
Simulate Responses of Model Nerve Fibers to Electrical Stimulation” describes a
well-developed Python package that models and simulates the activity of peripheral
nerve fibers within the NEURON simulation environment. It carries the overcoming of
several chronic obstacles in the field, such as the NEURON HOC language steep
learning curve proprietary barrier, reproducibility issues, and simulation fiber
over monolithic non-reusable code.

Recommendations for Improvement:

The mindset of continuously developing the software to address the gaps in usability
and standardization is admirable. However, the scientific and methodological novelty
does not seem to stand out in this particular case. The principal software
engineering achievements that provide the core value are contained within the
supporting algorithms and models, which have already been established in earlier
literature. Instead, consider rephrasing the framing for the software’s
contributions toward positioning it as enabling infrastructure that paves the way
for innovation and reproducibility rather than a novel scientific finding.

In my opinion, the manuscript would be improved by including a comparison of
performance metrics such as runtime, ease of use (lines of code), and accuracy
between PyFibers and the traditional HOC methods. Quantitative benchmarks for
various system configurations (e.g., with parallel simulations versus without) could
assist potential users in assessing the system’s applicability to their research
needs.

The manuscript could be enhanced by incorporating more PyFibers use cases in
realistic experimental workflows such as incorporating data from electrode arrays
and comparing it to electrophysiological recordings. Even a small illustrative
example of alignment between PyFibers simulations and actual recordings like SFAPs
would demonstrate the utility and impact of PyFibers in simulation.

Other tools like ASCENT or those built around HOC are mentioned, but a more thorough
discussion would be helpful to identify relative benefits of PyFibers.

Several hyperlinks (to documentation, GitHub, or ReadTheDocs) are listed as
placeholders (e.g. <

>) which disrupt narrative flow. These should be finalized to ensure reviewers and
prospective users will navigate freely throughout the submission.

The biologically unrealistic parameters such as fiber diameters or absent mechanisms
would be drastic user errors for PyFibers. The paper does not talk about how these
edge cases would be handled. It would be better with at least minimal mentions of
these gaps.

The choice of fiber models is an important one. Although it is nice to see models
from prior work being used, some reasoning behind the rationale for model selection
could be discussed. Will there be community-wide mechanisms for model submissions or
for extending the model library?

Additional overview schematics, not only based on the simulation flow diagrams, would
aid users in understanding the holistic interactions among fiber models, simulation
protocols, and the analysis tools and their workflows, thus aiding user
comprehension. The visualizations provided are clear and informative. Make sure that
high-resolution versions are included in the final submission. Specifically, within
figure legends, check for consistency in notation (e.g., Vm, vm, Vm(t)) and ensure
uniformity across the manuscript.

Overall evaluation: This work makes a valuable and innovative contribution to the
field of computational neuromodulation. It needs to be improved based upon the
comments above.

Reviewer #2: The authors present an open-source Python software library, "PyFibers",
designed to provide an efficient computational framework for simulating peripheral
nerve fiber responses to electrical stimulation. The library is built upon the
classic HOC-based NEURON environment and introduces a modular, extensible framework
that includes a baseline of 10 experimentally validated fiber models. The proposed
framework also features stimulation models, a user interface, documentation,
tutorials, and unit testing. However, at this stage, the code is not yet publicly
available.

The manuscript was submitted as a “software article” rather than a “research paper,”
which is the appropriate classification for this work, and it has been reviewed
accordingly.

Overall, I find the framework relevant and potentially impactful for computational
neuroscientists who aim to simulate morphologically driven phenomena. The proposed
system is well-structured, object-oriented, and cross-platform compatible, with
apparent integration into existing pipelines such as ASCENT. The package is
accompanied by usage guides and API documentation and includes support for both
myelinated and unmyelinated fiber models. Importantly, the authors validated their
implementations against previously published models.

Areas for Improvement

1. Benchmarking. The authors highlight a reduction in the lines of code required
(which is reasonable), but they should also consider evaluating their framework in
terms of quantitative performance metrics, particularly runtime efficiency.
Comparing PyFibers to traditional HOC implementations on complex, large-scale
morphological models would help demonstrate its practical usability and
scalability.

2. Functional Use Cases. A primary weakness of the study is the absence of
demonstrations involving realistic, functionally significant simulation scenarios.
While the authors validate their models against previously published work, they do
not showcase how PyFibers can be applied to novel, morphologically driven
investigations. While this limitation is somewhat acceptable given the
software-focused nature of the manuscript, a discussion on potential applications
would significantly enhance the paper’s impact. For instance, referencing studies
such as [https://doi.org/10.1371/journal.pcbi.1009754](https://doi.org/10.1371/journal.pcbi.1009754) and
discussing how PyFibers could support similar simulations would enrich the
discussion and emphasize the tool’s relevance.

**Have the authors made all data and (if applicable) computational code
underlying the findings in their manuscript fully available?**

Reviewer #1: Yes

Reviewer #2: Yes

PLOS authors have the option to publish the peer review history of their article
(what does this mean? ). If published, this will
include your full peer review and any attached files.

If you choose “no”, your identity will remain anonymous but your review may still be
made public.

**Do you want your identity to be public for this peer review?** For
information about this choice, including consent withdrawal, please see our
Privacy Policy .

Reviewer #1: No

Reviewer #2: No

**Figure resubmission:**
---

## [Decision Letter · Decision Letter 1]

11 Nov 2025

PCOMPBIOL-D-25-00791R1

PyFibers: An open-source NEURON-Python package to simulate responses of model nerve
fibers to electrical stimulation

PLOS Computational Biology

Dear Dr. Grill,

Thank you for submitting your manuscript to PLOS Computational Biology. After careful
consideration, we feel that it has merit but does not fully meet PLOS Computational
Biology's publication criteria as it currently stands. Therefore, we invite you to
submit a revised version of the manuscript that addresses the points raised during
the review process.

Please submit your revised manuscript by Jan 11 2026 11:59PM. If you will need more
time than this to complete your revisions, please reply to this message or contact
the journal office at ploscompbiol@plos.org. When you're ready to submit your
revision, log on to https://www.editorialmanager.com/pcompbiol/ and select the
'Submissions Needing Revision' folder to locate your manuscript file.

We look forward to receiving your revised manuscript.

Kind regards,

Alain Nogaret, PhD

Academic Editor

PLOS Computational Biology

Daniele Marinazzo

Section Editor

PLOS Computational Biology

**Additional Editor Comments:**

Please address the further suggestions for improvement namely:

- Improve the readability of the Design and Implementation section by making it more
succinct and accessible to non-programming readers.

- Briefly explain how future community contributions will be moderated or
version-controlled.

- Ensure the consistency in notations (e.g., Vm vs. Vₘ, μm vs. um) throughout the
figures and tables.

**Reviewers' comments:**

Reviewer's Responses to Questions

**Comments to the Authors:**

Reviewer #1: Compared to the original submission, this revised manuscript is
significantly better. The authors have now addressed all the previous criticisms by
the reviewers and editors in a concise and clear manner. In the revised version,
PyFibers is available as open-source, validated computational code, thus greatly
enhancing access, reproducibility, and standardization for simulations involving
nerve fibers.

Minor comments:

The Design and Implementation section could be tightened slightly to improve
readability, particularly for non-programming readers. It would also help to briefly
explain how future community contributions will be moderated or version-controlled.
Lastly, ensure consistency in notation (e.g., Vm vs. Vₘ, μm vs. um) throughout the
figures and tables.

**Have the authors made all data and (if applicable) computational code
underlying the findings in their manuscript fully available?**

Reviewer #1: Yes

PLOS authors have the option to publish the peer review history of their article
(what does this mean? ). If published, this will
include your full peer review and any attached files.

If you choose “no”, your identity will remain anonymous but your review may still be
made public.

**Do you want your identity to be public for this peer review?** For
information about this choice, including consent withdrawal, please see our
Privacy Policy .

Reviewer #1: No

**Figure resubmission:**
---

## [Editor Report · Decision Letter 2]

18 Nov 2025

Dear Dr Grill,

We are pleased to inform you that your manuscript 'PyFibers: An open-source
NEURON-Python package to simulate responses of model nerve fibers to electrical
stimulation' has been provisionally accepted for publication in PLOS Computational
Biology.

Best regards,

Alain Nogaret, PhD

Academic Editor

PLOS Computational Biology

Daniele Marinazzo

Section Editor

PLOS Computational Biology

Suggestions for minor improvements have been met in the revised version. The
manuscript is suitable for publication.

---

## [Editor Report · Acceptance letter]

PCOMPBIOL-D-25-00791R2

PyFibers: an open-source NEURON-Python package to simulate responses of model nerve
fibers to electrical stimulation

Dear Dr Grill,

I am pleased to inform you that your manuscript has been formally accepted for
publication in PLOS Computational Biology. Your manuscript is now with our
production department and you will be notified of the publication date in due
course.

With kind regards,

Anita Estes
